https://doi.org/10.1038/s41467-019-09289-5　　**OPEN**

# FBXW2 suppresses migration and invasion of lung cancer cells via promoting β-catenin ubiquitylation and degradation

Fei Yang[1,2], Jie Xu[2], Hua Li[2], Mingjia Tan[2], Xiufang Xiong[1] & Yi Sun[1,2]

FBXW2 inhibits proliferation of lung cancer cells by targeting SKP2 for degradation. Whether and how FBXW2 regulates tumor invasion and metastasis is previously unknown. Here, we report that FBXW2 is an E3 ligase for β-catenin. FBXW2 binds to β-catenin upon EGF-AKT1-mediated phosphorylation on Ser[552], and promotes its ubiquitylation and degradation. FBXW2 overexpression reduces β-catenin levels and protein half-life, whereas FBXW2 knockdown increases β-catenin levels, protein half-life and transcriptional activity. Functionally, FBXW2 overexpression inhibits migration and invasion by blocking transactivation of MMPs driven by β-catenin, whereas FXBW2 knockdown promotes migration, invasion and metastasis both in vitro and in vivo lung cancer models. In human lung cancer specimens, while FBXW2 levels are inversely correlated with β-catenin levels and lymph-node metastasis, lower FBXW2 coupled with higher β-catenin, predict a worse patient survival. Collectively, our study demonstrates that FBXW2 inhibits tumor migration, invasion and metastasis in lung cancer cells by targeting β-catenin for degradation.

[1] Cancer Institute of the Second Affiliated Hospital, and Institute of Translational Medicine, Zhejiang University School of Medicine, 310029 Hangzhou, China.
[2] Division of Radiation and Cancer Biology, Departments of Radiation Oncology, University of Michigan, 4424B MS-1, 1301 Catherine Street, Ann Arbor, MI MI48109, USA. Correspondence and requests for materials should be addressed to Y.S. (email: sunyi@umich.edu) or (email: yisun@zju.edu.cn)

L ung cancer, ~80% of which are non-small-cell lung cancer (NSCLC), is the leading cause of cancer-related deaths in the world[1]. Due to recurrence, extensive invasion and metastasis, the overall 5-year survival rate of NSCLC is lower than 15% after initial diagnosis[2]. Although multiple gene mutations, including EGFR, KRAS, p53, and PTEN, have been extensively reported in NSCLC[3] comprehensive molecular mechanisms that underlie the initiation, progression, and metastasis of NSCLC remain elusive. To pursue additional involving genes, we recently reported that FBXW2 (F-box and WD-repeat domain-containing 2), a poorly characterized F-box protein, acts as a tumor suppressor to inhibit growth and survival of lung cancer cells[4].

FBXW2, one of the 69 F-box proteins, functions as a substrate recognition receptor in the SCF (SKP1-Cullin1-F-box protein) ubiquitin ligase complexes[5]. The SCF ubiquitin ligases, also known as CRL1 (Cullin-RING ligase 1), consist of adapter protein SKP1, scaffold protein Cullin-1, Ring box protein-1 (RBX1)/ROC1, and an F-box protein. While Cullin and RING protein are required for ligase activity, the F-box protein determines the substrate specificity. As the largest family of ubiquitin ligases, SCF ubiquitin ligases promotes timely ubiquitylation and degradation of diverse regulatory proteins to control many biological processes[6,7]. FBXW2 is originally identified as an ubiquitin ligase for polyubiquitination and degradation of GCM1 (glial cell missing 1), which suppresses placental cell migration and invasion[8–10]. Our recent study identified FBXW2 as a tumor suppressor via promoting ubiquitylation of SKP2 (S phase kinase-associated protein 2) for targeted degradation to inhibit growth and survival of lung cancer cells[4]. However, whether FBXW2 targets other substrates to regulate the migration and invasion of lung cancer cells is totally unknown.

The Wnt/β-catenin pathway plays pivotal roles in development, cell proliferation, and differentiation[11]. β-catenin, an Armadillo protein, acts both as a component of cell–cell adhesion structure by interacting with the cytoplasmic domain of E-cadherin and as a cellular signaling molecule involved in the regulation of gene expression following Wnt pathway activation[12]. In the absence of Wnt (Wnt-off phase), cytoplasmic β-catenin forms a complex with axin/conductin, casein kinase 1 (CK1), glycogen synthase kinase-3β (GSK-3β), and the adenomatous polyposis coli protein (APC). CK1 and GSK-3β sequentially phosphorylate β-catenin at N-terminal Ser and Thr residues (Ser33, Ser37, Thr41, and Ser45), resulting in its ubiquitylation and proteasomal degradation by SCF$^{β-TrCP}$ ubiquitin ligase, in which F-box protein β-TrCP (β-transducin repeats-containing protein) acts as the substrate recognition receptor[13]. Upon exposure to the Wnt ligand (Wnt-on phase), the Axin complex is inactivated and hypophosphorylated β-catenin is stabilized by escaping from degradation, and then translocates to the nucleus, where it interacts with transcription factors, the TCF/LEF-1 family to transactivate the expression of Wnt response genes[14] including MYC[15], CCND1[16], MMP2[17], MMP7[18], and MMP9[17]. In various cancers, including colorectal, gastric, hepatocellular carcinoma, and NSCLC[19], β-catenin was found to be over-activated genetically or transiently, and accumulated in the nucleus of tumor cells. In addition, β-catenin can also be activated by Wnt-independent pathways, such as by growth factors including epidermal growth factor (EGF)[20,21], hepatocyte growth factor/scatter factor (HGF/SF)[22], and fibroblast growth factor (FGF)[23]. For example, EGF or HGF induced β-catenin activation to stimulate cell motility[20,21,24,25]. Upon EGF stimulation, β-catenin escaped from the degradation in cytoplasm and translocated into the nucleus to transactivate expression of genes involving in tumor invasion in a mechanism distinct from Wnt-dependent canonical signaling[20,21,24–26]. However, the ubiquitin ligase responsible for β-catenin degradation under this condition is little known.

Herein, we reported that the tumor suppressor FBXW2 is an ubiquitin ligase responsible for β-catenin degradation following EGF stimulation. Specifically, AKT1 (RAC-alpha serine/threonine-protein kinase 1) activation following EGF exposure phosphorylates β-catenin on Ser$^{552}$, which facilitates FBXW2 binding and subsequent ubiquitylation and degradation. Upon FBXW2 knockdown, accumulated β-catenin$^{pS552}$ enters the nucleus to transactivate expression of MMPs to promote migration, invasion and metastasis of lung cancer cells. Finally, we found that in human lung cancer tissues, FBXW2 levels are inversely correlated with β-catenin levels and lymph-node metastasis, and lower FBXW2, coupled with higher β-catenin, predicts a worse patient survival. Thus, FBXW2 suppresses migration, invasion, and metastasis of lung cancer cells via promoting ubiquitylation and degradation of β-catenin.

## Results

**FBXW2 binds to β-catenin via its consensus degron motif**. We have recently characterized FBXW2, a poorly characterized F-box protein, as a tumor suppressor that inhibited growth and survival of lung cancer cells by promoting ubiquitylation and degradation of oncogenic protein SKP2[4]. To further elucidate the mechanism of its anticancer activity, we used affinity purification-mass spectrometry method to identify FBXW2-binding proteins in H1299 cells with ectopic expression of tagged wild-type FBXW2. Among 196 putative binding proteins, we searched for consensus FBXW2 degron sequence T̲SXXX̲S, recently defined by us[4], and identified oncogenic protein β-catenin as a potential candidate, which contains such a motif (codons 551-556) in an evolutionarily conserved manner (Fig. 1a). Following this lead, we first confirmed the interaction between FBXW2 and β-catenin by plasmid co-transfection, followed by IP (immunoprecipitation) pull-down and IB (immunoblotting) detection, using β-TrCP, a known β-catenin binding protein[27–30] as positive control. Indeed, β-catenin can be readily detected in immunoprecipitants by FBXW2 or β-TrCP1, but not by FBXW7 or SKP2 (Fig. 1b). We next determined potential binding of endogenous β-catenin with a variety of F-box proteins and found that among 8 ectopically expressed F-box proteins (FBWX2, FBXW4, FBXW5, FBXW7, FBXW8, β-TrCP1, FBXL3, and FBXO4) only FBXW2 and β-TrCP1 bind with endogenous β-catenin (Fig. 1c). We further determined whether two proteins bind to each other under physiological conditions. Using IP pull-down assay in variety of human cell lines, including H1299, H358, MDA-MB231, HEK293, and HeLa cells, we detected endogenous β-catenin and SKP2 proteins in FBXW2 immunoprecipitants, and endogenous FBXW2 and β-TrCP1 in β-catenin immunoprecipitants, respectively. FBXL11, which was served as negative control, was not detected in either immunoprecipitants (Fig. 1d and Supplementary Figure 1a, b).

It is known that in most cases, phosphorylation is prerequisite for a substrate to bind to a F-box protein for targeted ubiquitylation and degradation by the SCF ubiquitin ligases[4,31]. To examine whether the interaction between FBXW2 and β-catenin is phosphorylation dependent, we constructed a panel of β-catenin mutants on FBXW2 degron motif, including phospho-mimetic (S552D) and phosphorylation-dead (S552A) single mutants, or triple mutants with all three S/T residues on the binding motif mutated (designated as β-catenin-3D or β-catenin-3A). Transfection followed by IP/IB assay revealed that compared to WT β-catenin constitutively active forms of β-catenin-S552D and β-catenin-3D bound more effectively to exogenous or endogenous FBXW2, whereas inactive forms of β-catenin-S552A and β-catenin-3A failed to do so (Fig. 1e, f). Together, these results clearly demonstrate an interaction

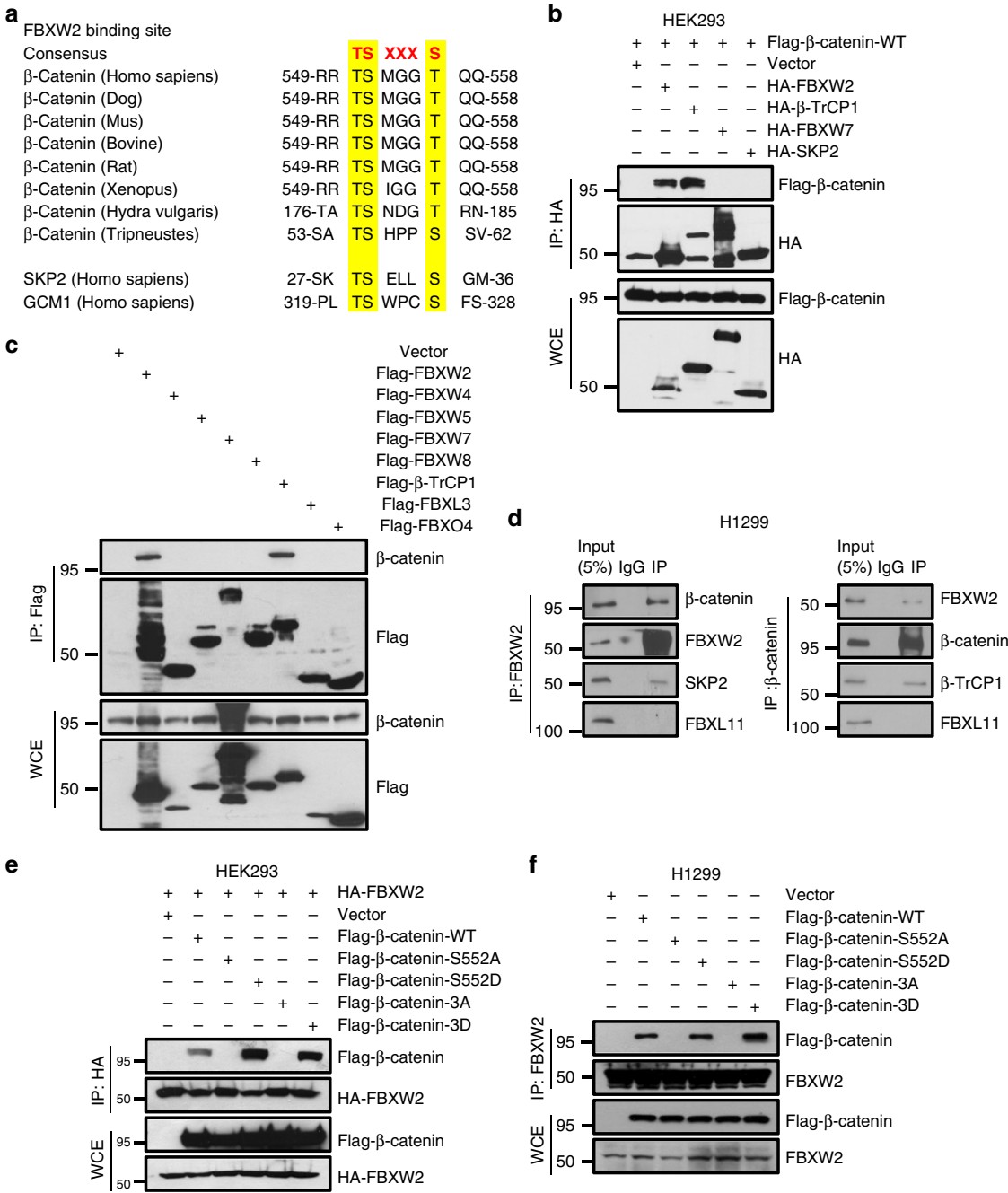

**Fig. 1** FBXW2 binds to β-catenin via its consensus degron motif. **a** Evolutionary conservation of FBXW2 degron motif on β-catenin. **b, c** FBXW2 binds to exogenously expressed (**b**) or endogenous β-catenin (**c**): HEK293 cells were transfected with indicated plasmids, followed by IP with anti-HA (**b**) or Flag Ab (**c**) and IB with indicated Abs. WCE: whole-cell extract. **d** FBXW2 binds to endogenous β-catenin: Cell lysates from H1299 cells were pulled down with anti-FBXW2 or anti-β-catenin Abs, followed by IB with indicated Abs. **e** The interaction between FBXW2 and β-catenin is dependent on the three S/T residues on the degron motif: β-catenin or its S→A or S→D mutants on FBXW2 degron motif was co-transfected with HA-FBXW2, followed by IP with HA Ab and IB with indicated Abs. **f** Endogenous FBXW2 binds to wild type β-catenin or its constitutively active mutants, but not its inactive mutants on the degron motif: H1299 cells were transfected with indicated plasmids, followed by IP with FBXW2 Ab and IB with indicated Abs. Unprocessed original scans of blots are shown in Supplementary Figure 9

between FBXW2 and β-catenin in a manner e/ssentially dependent of β-catenin-Ser$^{552}$ phosphorylation on the degron motif, and suggest that β-catenin is likely a previously unidentified substrate of FBXW2.

**FBXW2 reduces β-catenin levels and transcriptional activity.**
Having detected a physical interaction between FBXW2 and

β-catenin, we next assessed the effect of FBXW2 on β-catenin levels. We first observed in general an inversely correlated expression pattern between FBXW2 and phosphorylated β-catenin-Ser$^{552}$ as well as total β-catenin in multiple lung cancer cell lines (Supplementary Figure 2a). We then determined whether β-catenin level was negatively regulated by FBXW2. Indeed, while FBXW2 knockdown increased the levels of total β-catenin as well as phosphorylated β-catenin-Ser$^{552}$, FBXW2 overexpression

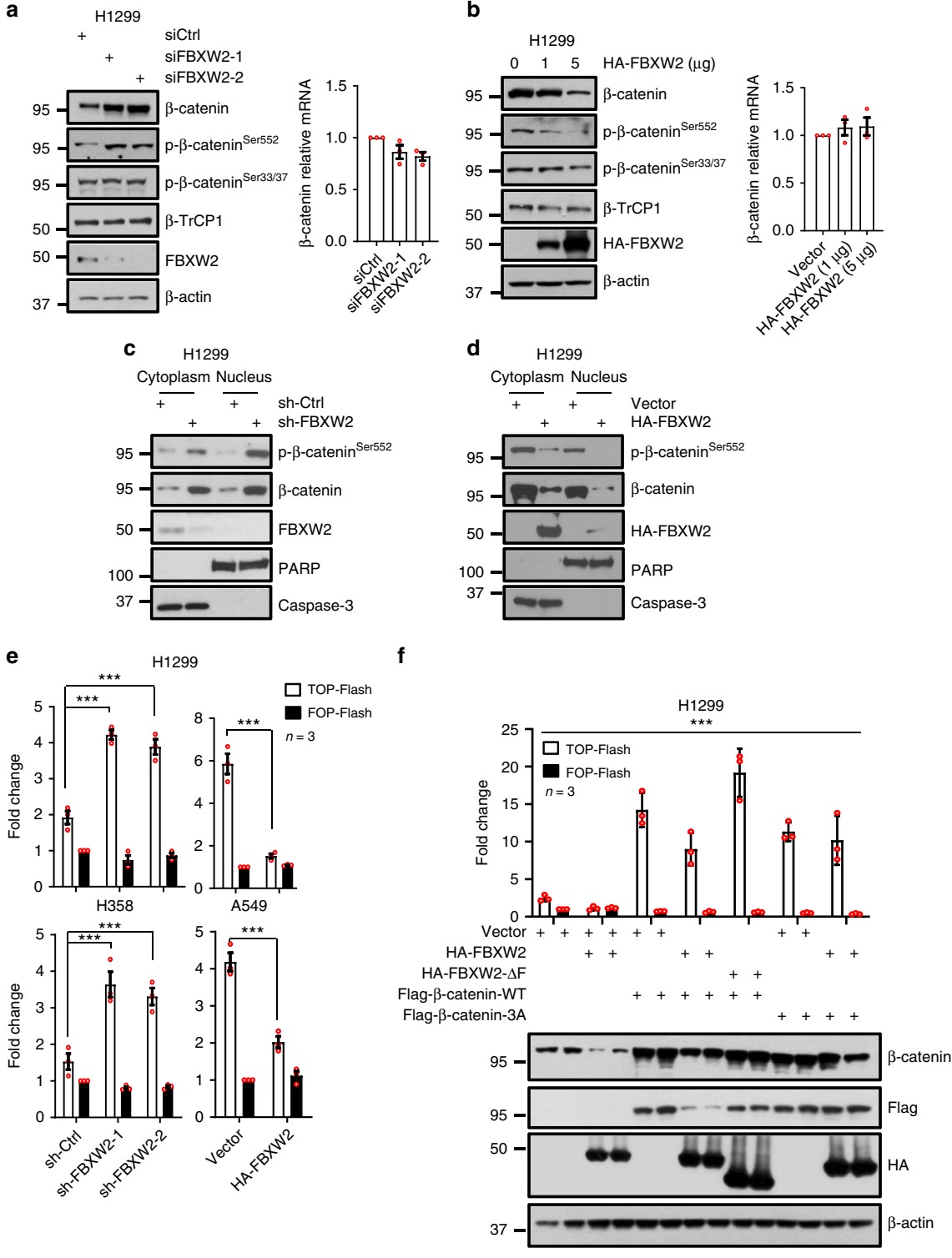

reduced them in a dose-dependent manner in various lung cancer cell lines. However, the levels of phosphorylated β-catenin-Ser$^{33/37}$ and total β-catenin mRNA were not affected by FBXW2 manipulations (Fig. 2a, b and Supplementary Figure 2b, c). Thus, FBXW2-induced reduction of β-catenin protein levels is through a post-transcriptional mechanism in a manner independent of phosphorylation of β-catenin on Ser$^{33/37}$ sites. On the other hand, β-TrCP1 manipulations negatively regulated protein levels of endogenous total β-catenin, phosphorylated β-catenin-Ser33/37 as well as FBXW2, but positively regulated phosphorylated β-

catenin-Ser552. These results are consistent with our recent findings that β-TrCP1 negatively regulated FBXW2 by promoting its ubiquitylation and degradation[4].

It is well-established that endogenous β-catenin undergoes a cytoplasm-to-nucleus shuttling and the hallmark of β-catenin activation is its stabilization and nuclear translocation[14], we then analyzed the cytosolic and nuclear quantities of endogenous β-catenin in FBXW2-manipulated cells. FBXW2 is mainly a cytoplasmic protein. Compared to that in control cells, the amount of β-catenin in both cytoplasmic and nuclear fractions

**Fig. 2** FBXW2 reduces β-catenin levels and transcriptional activity. **a** FBXW2 knockdown increases the endogenous protein levels of total β-catenin as well as phosphorylated β-catenin-Ser[552], but not phosphorylated β-catenin-Ser[33/37], nor β-catenin mRNA levels: H1299 cells were transfected with siRNA oligonucleotide targeting FBXW2, followed by IB (left) or qRT-PCR analysis (right). Data shown are mean±s.e.m of three independent experiments. **b** FBXW2 overexpression reduces phosphorylated β-catenin-Ser[552] and total β-catenin protein levels, but not β-catenin-Ser[33/37], nor β-catenin mRNA levels: H1299 cells were transfected with increasing amounts of FBXW2, followed by IB with indicated Abs (left) or qRT-PCR analysis (right). Data shown are mean±s.e.m of three independent experiments. **c** FBXW2 silencing increases β-catenin protein levels in both cytoplasmic and nuclear fractions: H1299 cells were infected with lentivirus expressing shRNA targeting FBXW2 or scrambled control shRNA (sh-Ctrl), followed by nuclear fractionation and immunoblotting. Cytoplasmic caspase-3 and nuclear PARP were used to determine the purity of each fractionation. **d** FBXW2 overexpression reduces β-catenin protein levels in both cytoplasmic and nuclear fractions: H1299 cells were transfected with FBXW2 or mock vector, followed by nuclear fractionation and IB with indicated Abs. **e** FBXW2 regulates the transcriptional activity of β-catenin: H1299 and H358 cells infected with lentivirus expressing shRNA targeting FBXW2 or scrambled control shRNA, or H1299 and A549 cells transfected with FBXW2 or mock vector were transfected with TOP-Flash with TCF-responsive promoter reporter or FOP-Flash with non-responsive control reporter. Transfection efficiency was normalized by co-transfection with pRL-TK. Luciferase activity was measured 48 h post transfection by the dual-luciferase assay. Data shown are mean±s.e.m of three independent experiments. *** p<0.001 (Student's t test). **f** Overexpression of FBXW2, but not FBXW2-ΔF, blocks the transcriptional activity of wild-type β-catenin: H1299 cells were transfected with wild-type β-catenin (WT) or β-catenin-3A, in combination with FBXW2-WT or FBXW2-ΔF, followed by TCF/β-catenin reporter dual-luciferase assay (top) and IB with indicated Abs (bottom). Data shown are mean±s.e.m of three independent experiments, ***p < 0.001 (One-way ANOWA). Unprocessed original scans of blots are shown in Supplementary Figure 9

was substantially higher in FBXW2 knockdown cells, whereas the amount of β-catenin in both fractions was significantly reduced in FBXW2-transfected cells (Fig. 2c, d). These results imply that increased nuclear β-catenin in FBXW2 knockdown cells is likely due to increased cytoplasmic β-catenin which is shuttled to the nucleus.

To evaluate whether increased nuclear β-catenin is transcriptionally active, we used paired TOP-Flash (which is subjected to T-cell factor/β-catenin transactivation) and FOP-Flash control luciferase reporters and found that β-catenin transactivation activity was increased upon FBXW2 knockdown, but reduced upon FBXW2 overexpression in multiple lines of lung cancer cells (Fig. 2e). Furthermore, the transcriptional activity of β-catenin was blocked by wild type FBXW2, but not by FBXW2-ΔF, a dominant-negative mutant that binds to the substrates but fails to recruit other components of SCF ubiquitin ligase[4] (Fig. 2f). Finally, the transcriptional activity of β-catenin-WT, but not β-catenin-3A, which failed to bind to FBXW2, was significantly suppressed by FBXW2 overexpression (Fig. 2f). Taken together, these results demonstrate that FBXW2 reduced the cytoplasmic β-catenin, leading to inhibition of its transcriptional activity in the nucleus.

**FBXW2 ubiquitylates β-catenin and shortens its half-life**. To further test whether FBXW2 regulates the stability of β-catenin, we next determined the protein half-life of β-catenin upon manipulation of FBXW2 in the presence of cyclohexamide (CHX) to block new protein synthesis. FBXW2 knockdown in lung cancer cells or *Fbxw2* depletion in MEF cells remarkably extended the protein half-lives of both phospho-β-catenin[Ser552] and total β-catenin (Fig. 3a, b and Supplementary Figure 3a), whereas FBXW2 ectopic expression significantly shortened them, but had no effect on phospho-β-catenin[Ser33/37] (Fig. 3c and Supplementary Figure 3b). Moreover, overexpression of FBXW2 shortened the protein half-life of ectopically expressed WT β-catenin (β-catenin-WT), but not that of β-catenin-3A mutant (Supplementary Figure 3c), indicating that the stability of β-catenin, negatively regulated by FBXW2, is dependent on the FBXW2 degron motif. Consistently, β-catenin-S552A is more stable in EGF stimulated cells as compared to β-catenin-WT/S552D/S33Y (Supplementary Figure 3e).

We next investigated, using classic in vivo and in vitro ubiquitylation assays, whether FBXW2 indeed promoted the ubiquitylation of β-catenin for enhanced degradation. The in vivo ubiquitylation assay showed that like β-TrCP, wild-type FBXW2, but not FBXW2-ΔF, significantly promoted polyubiquitylation of

exogenous β-catenin (Supplementary Figure 3d). Wild-type FBXW2 also promoted polyubiquitylation of endogenous β-catenin and this activity was remarkably reduced when FBXW2-ΔF mutant was used (Fig. 3d). Furthermore, FBXW2 only promoted polyubiquitylation of wild-type β-catenin and phosphorylation-mimicking mutant β-catenin-3D, but not the phosphorylation-dead mutant β-catenin-3A (Fig. 3e). Similarly, the in vitro polyubiquitylation assay, using an in vitro purified system containing E1, E2, E3s (FBXW2, FBXW2-ΔF) and substrates (β-catenin-WT, β-catenin-3A, β-catenin-3D), showed that FBXW2, but not its ΔF mutant, significantly promoted polyubiquitylation of β-catenin-WT and β-catenin-3D, but had no effects on β-catenin-3A (Fig. 3f). Finally, using various ubiquitin mutants, we showed that FBXW2-mediated β-catenin polyubiquitylation is via the K48 linkage for targeted degradation (Fig. 3g). Taken together, we concluded that FBXW2 acts as an ubiquitin ligase that promotes ubiquitylation and subsequent degradation of β-catenin to shorten its protein half-life.

**AKT1 is required for FBXW2-mediated β-catenin degradation.** It has been previously reported that AKT1, activated by EGF, phosphorylates β-catenin at the Ser[552] to promote its accumulation in the nucleus[21]. To investigate whether AKT1 is responsible for FBXW2-mediated β-catenin degradation, we first examined whether the FBXW2-β-catenin binding is affected by AKT1 manipulation. Indeed, the FBXW2-β-catenin binding was abrogated by siRNA-based AKT1 knockdown (Fig. 4a), but enhanced by EGF treatment, which was reversed by AKT1 inhibitor MK2206 (Fig. 4b and Supplementary Figure 4a). We next determined whether AKT1 affected FBXW2-mediated β-catenin degradation and found that AKT1 inactivation by MK2206 reduced the level of phospho-β-catenin, but increased the level of β-catenin (Fig. 4c). Moreover, inactivation of AKT1 via siAKT1 or AKT1 inhibitor MK2206 extended β-catenin protein half-life (Fig. 4d, e, and supplementary Figure 4b), whereas overexpression of AKT1 or constitutively active AKT1 (CA-AKT1) enhanced β-catenin polyubiquitylation, which were reversed by MK2206 (Fig. 4f). Finally, proteasome inhibitor MG132 caused accumulation of phospho-β-catenin[S552], triggered by EGF treatment, which was completely abrogated by AKT1 inhibitor MK2206 (Fig. 4g, and supplementary Figure 4c). Taken together, our results indicate that AKT1 is responsible for β-catenin phosphorylation on Ser552, which facilitated its FBXW2 binding for subsequent ubiquitylation and degradation.

It has been well established that in the canonical Wnt pathway, β-catenin stability is controlled by GSK-3β-mediated

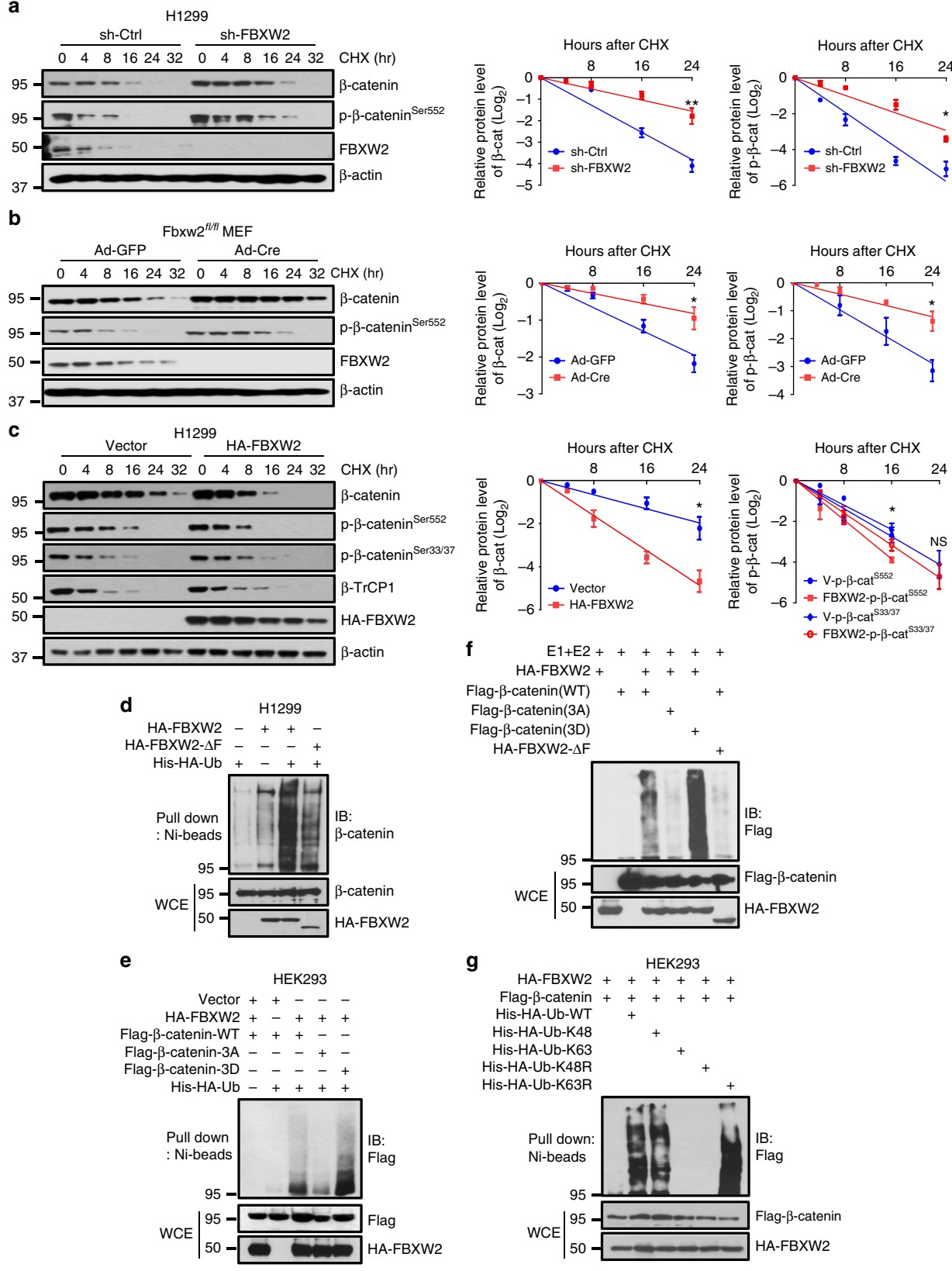

phosphorylation and subsequent β-TrCP1 binding for targeted ubiquitylation and degradation[27–30]. We next directly compared signals that trigger β-catenin binding with FBXW2 vs. with β-TrCP1. The β-catenin-FBXW2 interaction was enhanced by EGF treatment, completely blocked by siAKT1, but was not affected by Wnt3a treatment, nor by GSK-3β knockdown. On the other hand, β-catenin-β-TrCP1 interaction was inhibited by Wnt3a treatment, or GSK-3β knockdown, but was not affected by EGF or siAKT1 (Fig. 4h and Supplementary Figure 4d). Furthermore, β-catenin-S33Y, a β-catenin mutant, insensitive to

**Fig. 3** FBXW2 ubiquitylates β-catenin and shortens its half-life. **a**, **b** FBXW2 silencing extends the half-lives of both phospho-β-catenin and total β-catenin: H1299 cells were infected with lentivirus expressing shRNA targeting FBXW2 or scrambled control (**a**). Fbxw2$^{fl/fl}$ MEF cells were infected with Ad-Cre or Ad-GFP as control (**b**). Cells were then treated with cyclohexamide (CHX) for indicated time periods, followed by IB assay. **c** Overexpression of FBXW2 shortens the protein half-lives of both phospho-β-catenin-Ser$^{552}$ and total β-catenin, but has no effect on phospho-β-catenin-Ser$^{33/37}$: H1299 cells were transfected with HA-FBXW2 or mock vector for 48 h. Cells were then treated with CHX for indicated time periods before being harvested for IB assay (left panel). The band density was quantified using ImageJ software, normalized to β-actin first, then normalized to the $t = 0$ time point. Data are represented as mean±s.e.m. of three independent experiments, and *$p < 0.05$, **$p < 0.01$, NS, no significant difference (Student's $t$-test). **d** FBXW2 promotes β-catenin ubiquitylation in vivo: IB analysis of His tag pull-down and WCEs derived from H1299 cells transfected with indicated constructs. **e** FBXW2 promotes polyubiquitylation of wild-type β-catenin and its phosphorylation-mimicking mutant, but not the phosphorylation-dead mutant: IB analysis of His tag pull-down and WCEs derived from HEK293 cells transfected with indicated constructs. **f** FBXW2, but not its ΔF mutant, promotes polyubiquitylation of β-catenin-WT and β-catenin-3D, but has no effects on β-catenin-3A in vitro: FBXW2 (E3) was purified from HEK293 cells transfected with HA-FBXW2-WT or HA-FBXW2-ΔF by IP with HA beads and elution with 3×HA peptide. β-catenin and β-catenin mutants were prepared from HEK293 cells transfected with different β-catenin construct by IP with FLAG beads. FBXW2 (E3) and β-catenin (substrate) on FLAG beads were added into a reaction mixture containing ATP, ubiquitin, E1 and E2, followed by constant mixing at 37 °C for 60 min. The reaction mixture was then loaded onto SDS-PAGE for IB using anti-FLAG Ab. **g** FBXW2 promotes β-catenin ubiquitylation via K48 linkage: IB analysis of His tag pull-down and WCEs derived from HEK293 cells transfected with indicated constructs. Unprocessed original scans of blots are shown in Supplementary Figure 9

GSK-3β-mediated phosphorylation and β-TrCP1-mediated degradation[32], also bound to FBXW2, and the binding was blocked by AKT1 inhibitor MK2206 (Supplementary Figure 4e). Furthermore, while FBXW2 failed to bind to two β-catenin mutants (β-catenin-S552A, β-catenin-3A), β-TrCP1 bound to them equally well as to WT β-catenin (Fig. 1f and Supplementary Figure 4f). These results clearly demonstrated that FBXW2-mediated β-catenin degradation is independent of GSK-3β/β-TrCP1, but dependent on AKT1, whereas β-TrCP1-mediated β-catenin degradation is independent of AKT1/FBXW2, but dependent on Wnt3a/GSK-3β.

To further confirm this point, we measured the half-life of β-catenin S33Y and β-catenin-S552A when co-transfected with FBXW2 or β-TrCP1, respectively, and found that FBXW2 shortened the half-life of β-catenin-WT and β-catenin-S33Y, but not β-catenin-S552A, while β-TrCP1 shortened the half-life of β-catenin-WT and β-catenin-S552A, but not that of β-catenin-S33Y (Fig. 4i, j and Supplementary Figure 4g, h). Finally, FBXW2 inhibits the transcriptional activity of β-catenin activated by EGF, which is abrogated by AKT1 inhibitor, but has no effect on Wnt3a-induced β-catenin transactivation (Fig. 4k). Taken together, our results demonstrated that both the AKT1/FBXW2 and GSK-3β/β-TrCP1 axes can promote ubiquitylation and degradation of β-catenin, following phosphorylation at the residues S552 and S33, respectively.

**FBXW2 suppresses migration and invasion in vitro and in vivo.** To determine the biological significance of FBXW2-mediated β-catenin degradation, we measured effect of FBXW2 on migration and invasion of lung cancer cells, since it was reported that phosphorylation of β-catenin by AKT1 increased its transcriptional activity and promoted cancer cell invasion[20,21]. We transfected FBXW2 or β-catenin alone, or in combination into lung cancer cells and found that ectopic expression of FBXW2 suppressed cell migration and invasion, which can be blocked by simultaneous transfection of β-catenin. This blockage can be completely abrogated when cells were treated with GM6001, a small molecule inhibitor of MMPs[33] (see below; Fig. 5a and Supplementary Figure 5a). The results suggested that inhibitory effect of FBXW2 on migration and invasion of lung cancer cells is mediated mainly by targeted degradation of β-catenin. Consistently, FBXW2 depletion stimulated cell migration and invasion, which can be abrogated by simultaneous β-catenin depletion or GM6001 treatment as well as knockdown of MMP2 or MMP9 (Fig. 5b and Supplementary Figure 5b, 6a, 6b). Furthermore, we used porcupine inhibitor LGK974, which prevents secretion of all Wnt ligands[34], to evaluate the role of Wnt/β-

catenin signaling in cell migration stimulated by FBXW2 knockdown. The results showed that LGK974 at 1 μM had very minimal, if any, effect, while at 5 μM, it showed ~50% of blockage cell migration and invasion stimulated by FBXW2 knockdown (Supplementary Figure 6c), suggesting that the Wnt-β-TrCP1-β-catenin axis, while mainly controls proliferation, also contributes to migration and invasion, whereas the AKT1-FBXW2-β-catenin axis is more specific for migration and invasion.

Finally, we evaluated FBXW2 effect on in vivo lung metastasis by mouse tail vein injection of H1299 or H358 cells with stable FBXW2 knockdown alone or in combination with stable β-catenin knockdown. Six or eight weeks post inoculation, mice were killed and the lungs were collected. Compared with the control groups, the number of lung metastasis nodules in FBXW2 knockdown group was significantly increased, which is fully rescued by simultaneous β-catenin knockdown (Fig. 5c and Supplementary Figure 5c). Taken together, by targeting β-catenin, FBXW2 acts as an inhibitory protein against migration, invasion and metastasis of lung cancer cells in both in vitro cell culture and in vivo mouse models, although the tail-vein injection of tumor cells is not a spontaneous metastasis model.

**β-catenin$^{S552}$ induces MMPs to mediate migration and invasion.** As a transcription co-activator, β-catenin regulates a variety of cellular processes, including cell proliferation (via CCND1[16], MYC[15]), stem cell fate determination (via ASCL2[35]), migration (via MMP2[17], MMP7[18], MMP9[17]), and angiogenesis (via VEGF[36]). These processes were abnormally activated due to β-catenin deregulation seen in multiple human cancers[11,37]. To further elucidate how the FBXW2-β-catenin axis regulates migration and invasion, and its mechanism of action, we transfected various β-catenin mutants into H1299 cells and found that β-catenin-S552D, a constitutive active form upon AKT1 activation, is the most effective in promoting migration and invasion (Fig. 6a). Consistently, β-catenin-S552D is also the most active form in transactivating the expression of invasion/metastasis-promoting genes encoding MMP2, MMP7, and MMP9, while β-catenin-S33Y is the most active form in transactivating the expression of proliferation-promoting genes, encoding c-Myc and Cyclin D1 (Fig. 6b). In agreement of these results, the expression of MMP2, MMP7, and MMP9 in lung cancer H1299 or H358 cells were either down- or up-regulated by FBXW2 over-expression or knockdown, respectively, which can be rescued by manipulation of β-catenin expression or using MMPs inhibitor GM6001, further supporting a causal role of MMPs (Fig. 6c and Supplementary Figure 7a). Finally, we used the CHIP-based promoter-binding assay and found that mutant β-catenin-S552D

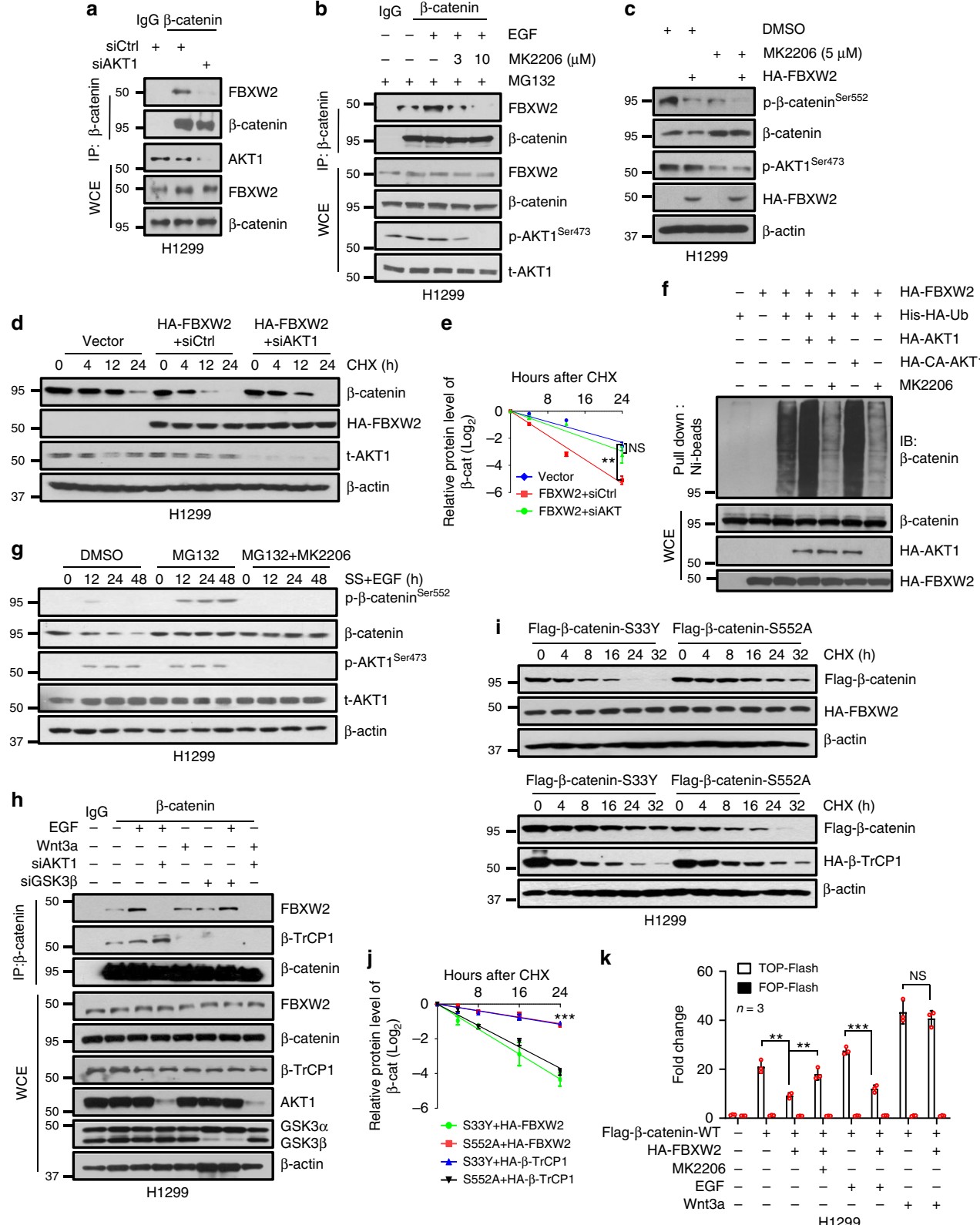

preferentially binds to the promoters of the *MMP-2, -7*, and *-9*, whereas mutant β-catenin-S33Y had preference to bind to the promoters of *MYC* and *CCND1* (Fig. 6d). Moreover, FBXW2 knockdown, which caused an increased level of β-catenin-S552D (Fig. 2a), facilitated β-catenin binding to the promoters of *MMP-2, -7*, and *-9*, whereas FBXW2 overexpression inhibited it (Fig. 6e and Supplementary Figure 7b). We then attempted to understand

the binding preference of β-catenin mutants to different gene promoters. It is well established that the downstream signaling events involving the Wnt/β-catenin cascade occur through T-cell factor (TCF)-4 transcription factor proteins[38,39], and different TCF4 isoforms generated by alternative splicing events activated different downstream target genes in hepatocellular carcinoma (HCC) and renal cell carcinoma (RCC)[40–42]. We, therefore,

**Fig. 4** AKT1 is required for FBXW2-mediated β-catenin degradation. **a** IB analysis of WCEs and IPs derived from H1299 cells transfected with indicated siRNA. **b** IB analysis of WCEs and IPs derived from H1299 cells pretreated with MK2206 for 2 h, followed by the treatment of EGF (100 ng/ml) and MG132 (10 μM) for additional 6 h. **c** H1299 cells transfected with FBXW2 or mock vector were left untreated or treated with MK2206 for 6 h, followed by IB. **d**, **e** H1299 cells were first transfected with indicated siRNA and constructs, then treated with CHX for indicated time periods before IB (**d**). The band density was quantified using ImageJ software. Data are represented as mean±s.e.m. of three independent experiments, and **p < 0.01, NS, no significant difference (Student's t test) (**e**). **f** IB analysis of His tag pull-down and WCEs derived from H1299 cells transfected with indicated constructs. CA-AKT1: constitutively active AKT1. **g** H1299 cells were serum starved for 24 h, followed by the addition of serum and EGF (100 ng/ml) in the absence or presence of MG132 (1 μM) and MK2206 (1 μM). Cells were then harvested at indicated time points for IB. **h** H1299 cells were transfected with indicated siRNA, followed by treatment with EGF (100 ng/ml) or Wnt3a (100 ng/ml) for 6 h. Cells were then treated with MG132 (10 μM) for the last 2 h before being harvested for IP with β-catenin Ab and IB. **i**, **j** H1299 cells were transfected with indicated plasmids, then treated with CHX for indicated time periods before being harvested for IB (**i**). The band density was quantified using ImageJ software. Data are represented as mean±s.e.m. of three independent experiments, ***p < 0.001 (One-way ANOWA) (**j**). **k** H1299 cells were transfected with indicated constructs, then treated with MK2206 (5 μM), EGF (100 ng/ml) or Wnt3a (100 ng/ml) for 6 h, followed by TCF/β-catenin reporter dual-luciferase assay. Data shown are mean±s.e.m of three independent experiments, and **p < 0.01, ***p < 0.001. NS no significant difference (Student's t-test). Unprocessed original scans of blots are shown in Supplementary Figure 9

performed the β-catenin pull-down assay after transfection of wild-type β-catenin and its few phospho-mutants to identify their TCF binding partners. Interestingly, we found that β-catenin-S33Y preferentially binds to TCF isoform TCF4E, whereas β-catenin-S552D preferred the isoform TCF4M/S (Supplementary Figure 7c). Taken together, these results demonstrate that FBXW2 inhibition of migration and invasion in lung cancer cells is mediated via promoting β-catenin degradation and consequent downregulation of MMP expression, which is preferentially regulated by the β-catenin-S552D-TCF4M/S transcription factors.

**The levels of FBXW2 and β-catenin predict patient survival**. To evaluate the clinical relevance of our findings, we measured FBXW2 and β-catenin expression patterns in lung cancer patients by immune-staining of a tumor tissue microarray. We found a general tendency of inversed protein levels between FBXW2 and β-catenin in the tumors (Fig. 7a). Quantification of the staining intensity showed that this inverse correlation was statistically significant among the 90 specimens analyzed ($r = -0.6861$; $p < 0.001$, Pearson correlation coefficient; Fig. 7b). Furthermore, we compared the expression levels of FBXW2 and β-catenin in the tumors with (51 cases) and without (39 cases) lymph-node metastasis, and found that the staining intensity of FBXW2 was significantly reduced in the tumors with lymph-node metastasis as compared to those without lymph-node metastasis, while the staining intensity of β-catenin was just opposite, which were statistically significant (Fig. 7c).

We have recently shown that FBXW2 expression is downregulated in lung cancer tissues and its downregulation correlated with poor survival of lung cancer patients ($n = 102$)[4]. To determine whether the inverse expression pattern between FBXW2 and β-catenin is correlated with patient overall survival, we performed the Kaplan–Meier survival analysis. Among 88 cancer samples (from a total of 180 lung cancer patients) with inverse correlation between FBXW2 and β-catenin, we found that high FBXW2/low β-catenin ($n = 20$) predicts a better patient survival, whereas low FBXW2/high β-catenin ($n = 68$) was associated with poor survival of lung cancer patients (Fig. 7d). Thus, inverse expression pattern between FBXW2 and β-catenin in lung cancer tissues indeed impacts the patient survival.

Finally, we performed immunostaining of β-catenin in xenograft tumors derived from H1299 stable clones with FBXW2 knockdown or overexpression[4]. The β-catenin levels were increased in the cytoplasm and nucleus in FBXW2 knock-down xenograft tumors, but decreased in wild type FBXW2-overexpressed tumors (Supplementary Figure 8). These association studies strongly suggested that the FBXW2-β-catenin axis

could play a critical role in regulation of migration and invasion/metastasis of human lung cancer.

## Discussion

The Wnt/Wingless-dependent or -independent signals stabilizes β-catenin and increases its nuclear translocation to act as a transcription coactivator to drive expression of genes controlling proliferation and invasion, while constitutive activation of β-catenin leads to tumorigenesis[11,14]. Given biological significance of β-catenin, it is not surprising that multiple ubiquitin ligases were found to regulate its stability, including β-TrCP[27–30], Siah-1[43], Jade-1[44], c-Cb1[45], and TRIM33[46]. However, ubiquitin ligase responsible for targeted ubiquitylation and degradation of β-catenin following EGF-AKT1 activation, leading to enhanced cell motility[21,25], is previously unknown.

Here we report that FBXW2 is the responsible E3 in this role. Our conclusion is supported by the following lines of evidence: (1) FBXW2 binds to β-catenin under physiological conditions, and FBXW2-β-catenin binding is dependent on an evolutionarily conserved FBXW2 consensus binding motif (TSMGGT) on β-catenin (Fig. 1); (2) FBXW2 ectopic expression or siRNA knockdown decreases or increases the levels of total and phosphorylated β-catenin (β-catenin$^{pS552}$), and β-catenin transcriptional activity, respectively (Fig. 2); (3) FBXW2 ectopic expression or siRNA knockdown shortens or extends the protein half-lives of both β-catenin and β-catenin$^{pS552}$, respectively, and FBXW2 promotes β-catenin polyubiquitylation via K48 linkage for degradation (Fig. 3); (4) EGF-activated AKT1 is responsible for β-catenin phosphorylation on Ser552, which facilitates its FBXW2 binding for degradation. Likewise, inactivation of AKT1 via genetic or small molecule approach extended β-catenin protein half-life (Fig. 4). Taken together, β-catenin can be degraded either in the absence of Wnt signal via GSK-3β-mediated phosphorylation at the N-terminus (Ser33Ser37) by SCF$^{β-TrCP}$ to regulate cell growth[27–30], or in response to activation of EGF signal via AKT1-mediated phosphorylation at the C-terminus (Ser552) by SCF$^{FBXW2}$ to regulate cell motility (this study and see below).

It is well-established that EGFR overexpression and abnormal activation is a dominant oncogenic signal for the development of many human malignancies including NSCLC, which is associated with reduced survival, frequent lymph node metastasis and poor chemo-radiation sensitivity[47–51]. Upon activation, EGFR activates the RAS-MEK-ERK and PI3K-AKT1 pathways that are central to the growth, survival, and migration of cancer cells[52,53]. It has been previously reported that AKT1-mediated β-catenin phosphorylation on Ser$^{552}$ increased its transcriptional activity and promoted tumor cell invasion[20]. Given that the residue Ser552 is within the FBXW2 consensus binding motif and that its

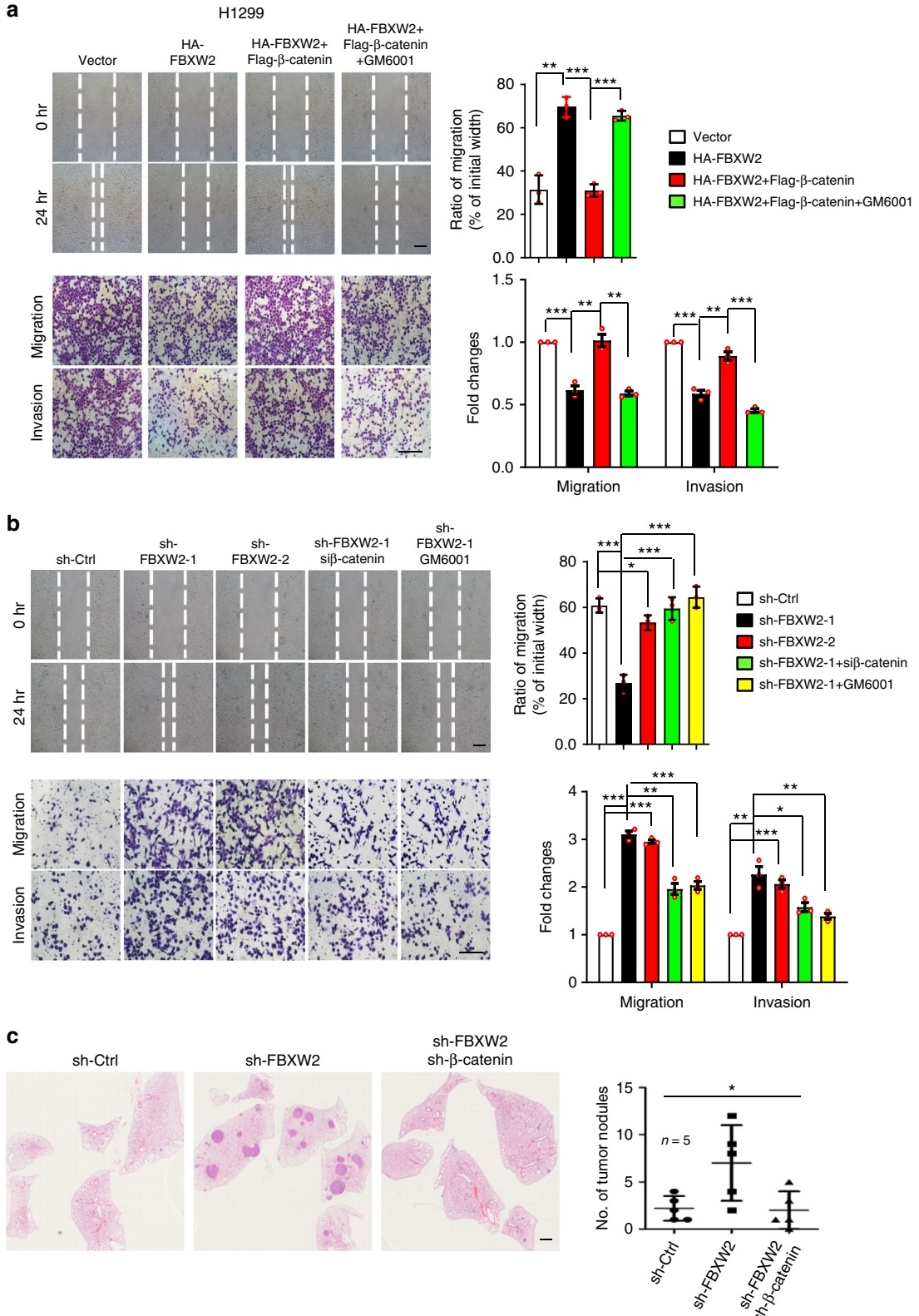

phosphorylation by AKT1 facilitated FBXW2 binding and subsequent β-catenin degradation, we, therefore, focused our functional studies on FBXW2 regulation of migration and invasion via targeting β-catenin upon phosphorylation by the EGF-AKT1 signal. Indeed, the results from several in vitro cell-based models, as well as an in vivo lung metastasis model via tail-vein injection, showed that FBXW2 negatively regulates migration, invasion, and

metastasis (Fig. 5). This negative effect appears to be in the manner dependent of β-catenin and its downstream targets MMPs, as evidenced by the rescue experiments with β-catenin manipulation, or with a small molecular inhibitor of MMPs or siRNA-based knockdowns of MMPs (Fig. 5, Supplementary Figs. 5 and 6). The lack of full rescue of FBXW2 effect by the Wnt signal inhibitor LGK974 (Supplementary Fig. 6) suggests that the

**Fig. 5** FBXW2 suppresses migration and invasion in vitro and in vivo. **a** Ectopic expression of FBXW2 suppresses cell migration and invasion, which can be blocked by simultaneous transfection of β-catenin in a MMPs-dependent manner: H1299 cells stably expressing indicated plasmids after G418 selection were treated with DMSO or GM6001 (5 μM), followed by wound-healing assay (top) and transwell chamber migration and invasion assays (bottom). **b** FBXW2 knockdown stimulates cell migration and invasion, which is abrogated by simultaneous silencing of β-catenin or GM6001 treatment: H1299 cells stably expressing shRNA targeting FBXW2 or scrambled control shRNA after puromycin selection were transfected with siRNA targeting β-catenin or scrambled control siRNA, or treated by DMSO or GM6001 (5 μM), followed by wound-healing assay (top) and transwell chamber migration and invasion assays (bottom). Shown are representative images of migrated cells (left panels). The ratios of migration were calculated by dividing the width of the wound after 24 h and the width of initial wound (top, right). The number of migrated cells was counted in three random fields per chamber and statistically analyzed (bottom, right). Data are shown as mean ± s.e.m of three independent experiments, *$p < 0.05$, **$p < 0.01$, ***$p < 0.001$ (Student's $t$-test). Scale bars, 100 μm. **c** FBXW2 knockdown stimulates metastasis of lung cancer cells in vivo: H1299 cells with stable FBXW2 knockdown alone or in combination with stable β-catenin knockdown were injected into nude mice via tail vein. After 6 weeks, mice were killed and lung tissues were stained with hematoxylin and eosin and photographed (left). The number of lung metastasis nodules in all five lobes of the lung from each mouse was counted and statistically analyzed (right). $n = 5$ for each group, *$p < 0.05$ (One-way ANOWA). Scale bars, 1 mm

AKT1-FBXW2 axis, rather than the Wnt-β-TrCP axis plays a major role in regulation of migration and invasion. The finding is consistent with our IHC study which showed (1) an inverse correlation between FBXW2 levels in lung cancer tissues and lymph-node metastasis potential, and (2) higher FBXW2, coupled with lower β-catenin, predicts a better patient survival, whereas lower FBXW2, coupled with higher β-catenin, predicts a worse patient survival (Fig. 7). These results are also consistent with our recent observation that lower FBXW2 levels are positively correlated with a poor patient survival[4].

It has also been reported that EGFR signaling downstream of EGF regulates migration and invasion through MMPs over-expression in malignant diseases[54–57], but the underlying mechanism remains elusive. Here we showed that the process is mediated by the AKT1-FBXW2-β-Catenin-MMPs axis (Figs. 5 and 6). Interestingly, we found that β-catenin[S33Y], a mutant resistant to GSK-3β-β-TrCP degradation, preferentially binds to the promoters of *MYC* and *CCND1*, whereas β-catenin[S552D], a constitutively activated form mimicking AKT1 phosphorylation, preferentially binds to the promoters of *MMPs*, which is likely mediated through selective binding to different TCF4 isoforms (Fig. 6, Supplementary Fig. 7). It has been previously reported that different TCF4 isoforms generated by alternative splicing events activated different downstream target genes in hepatocellular carcinoma (HCC) and renal cell carcinoma (RCC)[40–42]. Thus, the binding preference/difference observed here may explain why the Wnt-controlled β-TrCP-β-catenin axis mainly regulates cell proliferation, whereas the EGF-AKT1-controlled FBXW2-β-catenin axis mainly for invasion and metastasis. Our study, therefore, elucidated the signaling pathway that connects upstream EGF/AKT1 signal to downstream β-catenin[pS552]/MMPs in promoting invasion and metastasis, which is under precise surveillance by tumor suppressor FBXW2 via targeting β-catenin[pS552] for degradation.

In summary, our study reveals a complex interplay and cross-talk between the Wnt-GSK-3β and the EGF-AKT1 signals centered at downstream β-catenin, which is subject to targeted regulation by two F-box proteins, β-TrCP and FBXW2. Specifically, when β-catenin is differentially phosphorylated by different kinases in response to different signals (Fig. 4), there are two levels of cross-talks among β-TrCP, FBXW2, and β-catenin. First, both FBXW2 and β-catenin are the substrates of β-TrCP[4,27–30]. β-TrCP promotes FBXW2 ubiquitylation and degradation at the S phase upon FBXW2 phosphorylation by VRK2[4], whereas it promotes ubiquitylation and degradation of β-catenin upon GSK-3β-mediated phosphorylation, when the Wnt signal is off. Second, β-catenin is substrate of both FBXW2 and β-TrCP. Upon activation via EGF, AKT1, on one hand, inhibits GSK-3β[58,59] to prevent β-catenin phosphorylation at Ser[33], thus disrupting β-TrCP binding and degradation, and on the other hand,

phosphorylates β-catenin on Ser[552] to present it to FBXW2 for degradation. Thus, in lung cancer cells with low levels of FBXW2, non-phosphorylated β-catenin and phospho-β-catenin[S552] would accumulate in response to the EGF/AKT1 signal, followed by nuclear translocation to drive the expression of genes controlling both proliferation (e.g. c-Myc and Cyclin D1) and metastasis (e.g. MMPs) through partnering with different TCF4 isoforms (Fig. 7e). In this regards, FBXW2 plays a predominant role in regulating β-catenin signal upon EGF-AKT1 activation.

## Methods

**Cell culture**. Human embryonic kidney 293 (HEK293) cells, lung cancer H1299 and A549 cells, cervical carcinoma HeLa cells, and breast cancer MBA-MD-231 cells were maintained in Dulbecco's Modified Eagle's Medium (DMEM) containing 10% (v/v) fetal bovine serum (FBS). Lung cancer H358 cells were maintained in RPMI 1640 Medium with 10% FBS. ALL cell lines were maintained in Yi Sun lab. All cell lines were routinely tested to be negative for mycoplasma contamination.

**Plasmids, siRNAs, and reagents**. The plasmid constructs expressing HA-FBXW2, FLAG-FBXW2, HA-FBXW7, HA-β-TrCP1, HA-SKP2, FLAG-FBXW4, FLAG-FBXW5, FLAG-FBXW8, FLAG-β-TrCP1, FLAG-FBXL3, and FLAG-FBXO4 were maintained in Yi Sun lab. TOP-Flash, FOP-Flash, and pRL-TK were obtained from Dr. Li Ma (The University of Texas M.D. Anderson Cancer Center, Houston, Texas, USA). HA-AKT1 and HA-CA-AKT1 were obtained from Dr. Wenyi Wei (Harvard Medical School, Boston, MA, USA). Human β-catenin and β-catenin mutants were generated by PCR and using the QuikChange Site-Directed mutagenesus kit (Agilent Technologies, #200522), according to the manufacturer's instructions. The primers used to generate the β-catenin mutants are shown in Supplementary Table 1.

The pooled siRNA oligos targeting for β-catenin, AKT1, and GSK-3β were obtained from Santa Cruz. The sequences of siRNA oligos and siRNA oligos used for construction of lentivirus silencing vector are shown in Supplementary Table 2.

Recombinant human Wnt3a and recombinant human EGF were purchased from R&D Systems. AKT1 inhibitors MK2206 and MMPs inhibitors GM6001 were purchased from Selleck. Wnt3a inhibitor LGK974 was purchased from MedChemExpress.

**Transfection and lentiviral infection**. Cells were transfected with a variety of plasmids or siRNA oligos, using Lipofectamine 2000, according to the manufacturer's protocols, followed by various assays 48 h post transfection. For gene knockdown, cells were infected with the lentivirus expressing sh-RNAs, produced by University of Michigan Vector Core, and then selected with 1.5 μg/ml puromycin for stable cell lines, or followed by various assays 96 h post infection.

**Immunoprecipitation and immunoblotting**. Cells were lysed in an IP lysis buffer (50 mM Tris-HCl, pH 8.0, 120 mM NaCl, 0.5% NP40, 1 mM EDTA), supplemented with complete protease inhibitor cocktail (Complete Mini, Roche). For immunoprecipitation, 1–2 mg lysates were incubated with bead-conjugated FLAG or the appropriate antibody (2 μg) in a rotating incubator overnight at 4 °C, followed by 2 h incubation with Protein-A Sepharose beads (GE Healthcare). Immuno-complexes were washed four times with IP lysis buffer before resolved by SDS-PAGE, and analyzed by immunoblot. For direct IB analysis, cells were lysed in lysis buffer (50 mM Tris-HCl, pH 8.0, 150 mM NaCl, 1% TritonX-100, 1% sodium deoxycholate, and 0.1% SDS, 1 mM EDTA). Proteins were resolved on SDS polyacrylamide gels, and then transferred to a polyvinylidene difluoride membrane. After blocking with 5% (w/v) milk, the membrane was stained with the corresponding primary antibodies.

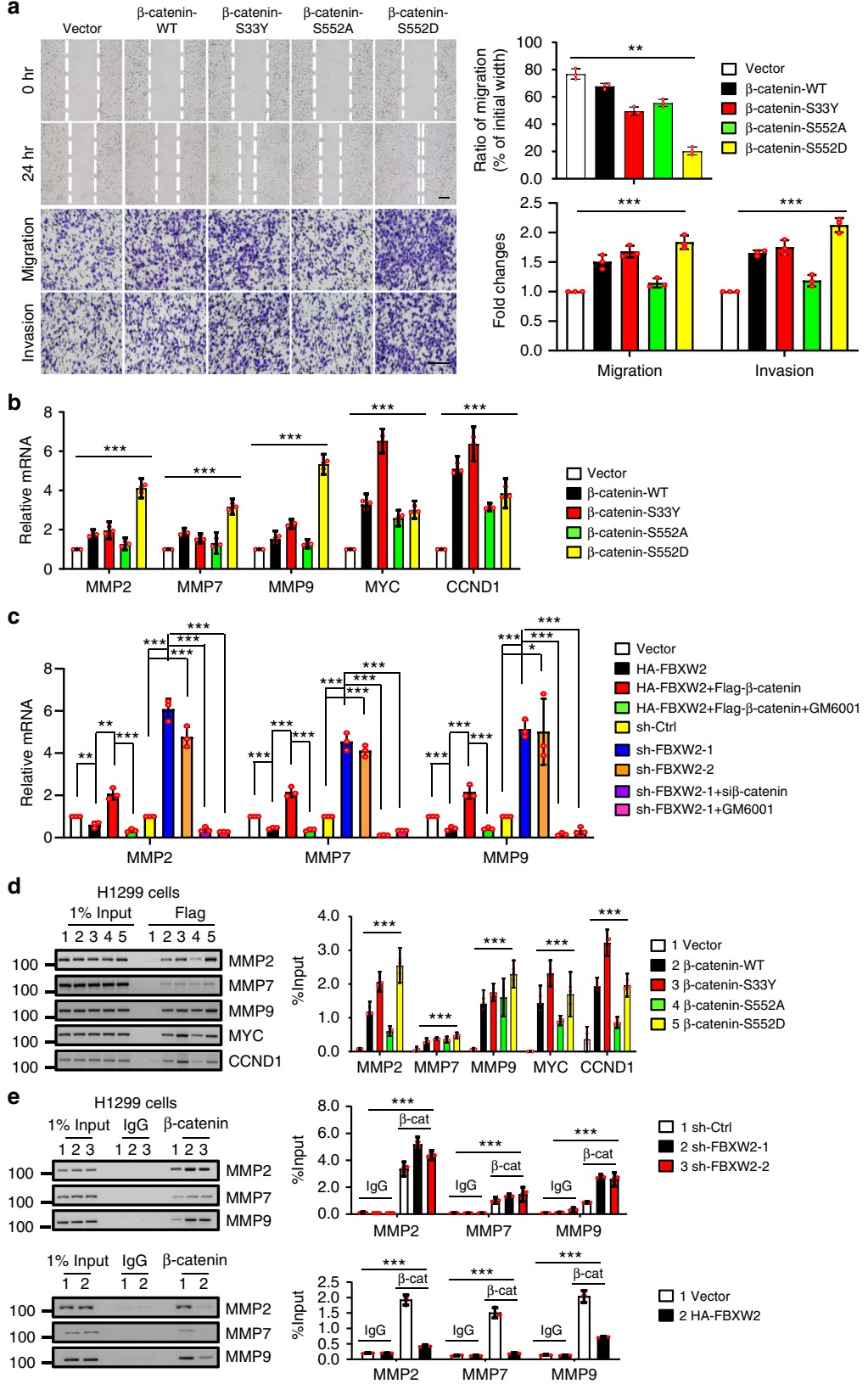

The antibodies were purchased as follows: β-catenin (Santa Cruz, sc-59737, 1:1000 for immunoblotting, 1:400 for immunohistochemistry), FBXW2 (Abcam, ab83467, 1:1000 for immunoblotting; Proteintech, #11499-1-AP, 1:1000 for immunoblotting, 1:100 for immunohistochemistry). The following antibodies were used for immunoblotting only: phospho-β-catenin (Ser552) (Cell Signaling, #5651,1:1000), SKP2 (Cell Signaling, #2652, 1:1000), β-TrCP1 (Cell Signaling,#4394, 1:1000), HA (Roche Life Science, #11867423001, 1:1000), FLAG (Sigma, #F1804, 1:1000), β-Actin (Santa Cruz, sc-47778, 1:1000), FBXL11 (Santa Cruz, sc-135126, 1:1000), GSK-3β (Cell Signaling, #12456, 1:1000), AKT1 (Cell Signaling, #4691, 1:1000), phospho-AKT1 (Ser473) (Cell Signaling, #4051, 1:1000), PARP (Cell Signaling, #9532, 1:1000), and caspase3 (Santa Cruz, sc-7272, 1:1000), phospho-β-catenin (Ser33/37Thr41) (Cell Signaling, #9561, 1:1000), TCF4 (Cell Signaling, #2569, 1:1000).

**Fig. 6** β-catenin$^{S552}$ induces MMPs to mediate migration and invasion. **a** β-catenin-S552D is the most effective form to promote migration and invasion: H1299 cells stably expressing indicated plasmids after G418 selection were subjected to wound-healing assay (top) and transwell chamber migration and invasion assays (bottom). Shown are representative images of migrated cells (left panels). The ratios of migration were calculated by dividing the width of the wound after 24 h and the width of initial wound (top, right). The number of migrated cells was counted in three random fields per chamber and statistically analyzed (bottom, right). Data are shown as mean±s.e.m. of three independent experiments, **$p < 0.01$, ***$p < 0.001$ (One-way ANOWA). Scale bars, 100 μm. **b** β-catenin-S552D and β-catenin-S33Y are the most active forms to transactivate the expression of metastasis-promoting genes and proliferation-promoting genes, respectively: H1299 cells were transfected with indicated plasmids, followed by qRT-PCR analysis to detect the mRNA expression of β-catenin downstream targets as indicated. Data are shown as mean±s.e.m. of three independent experiments, ***$p < 0.001$ (One-way ANOWA). **c** FBXW2 negatively regulates the expression of MMPs, which is rescued by manipulation of β-catenin expression or using GM6001: H1299 cells transfected with indicated plasmids or infected with indicated lentivirus were left untreated or treated with GM6001, followed by qRT-PCR analysis. Data are shown as mean±s.e.m. of three independent experiments, *$p < 0.05$, **$p < 0.01$, ***$p < 0.001$ (Student's $t$-test). **d** β-catenin-S552D and β-catenin-S33Y preferentially bind to the promoters of *MMP* genes, *MYC* and *CCND1*, respectively: H1299 cells were transfected with indicated Flag-β-catenin constructs, followed by ChIP assays with anti-Flag antibody. Data are shown as mean±s.e.m. of three independent experiments, ***$p < 0.001$ (One-way ANOWA). **e** FBXW2 negatively regulates β-catenin binding to the promoters of *MMP* genes: H1299 cells were infected with indicated lentiviruses or transfected with indicated plasmids, followed by ChIP assays with anti-β-catenin antibody. Mouse IgG was used as a negative control. Data are shown as mean±s.e.m. of three independent experiments, ***$p < 0.001$(One-way ANOWA). Unprocessed original scans of gel source images are shown in Supplementary Figure 9

**Subcellular fractionation**. For nuclear and cytoplasmic fractionations, cells were washed twice with cold PBS and collected by scraping, followed by the addition of lysis buffer A (20 mM HEPES, pH 8.0, 20% glycerol,10 mM NaCl, 1.5 mM MgCl2, 0.2 mM EDTA, pH 8.0, 1 mM DTT, 0.1% NP-40, and a proteinase inhibitor cocktail; Roche). Cell lysates were incubated on ice for 10 min, followed by centrifugation at $400 \times g$ for 5 min at 4 °C. The supernatant was transferred (cytoplasmic extract) to a clean pre-chilled tube. The pellet was sequentially washed 3 times with lysis buffer A and then lysed in buffer B (20 mM HEPES, pH 8.0, 20% glycerol, 500 mM NaCl, 1.5 mM MgCl$_2$, 0.2 mM EDTA, pH 8.0, 1 mM DTT, 0.1% NP-40 and a proteinase inhibitor cocktail) for 30 min on ice, followed by centrifugation at $15,000 \times g$ at 4 °C for 15 min. The supernatants were collected as the nuclear fractions.

**Dual-luciferase assay**. Cells were transfected with the TOP-FLASH or FOP-FLASH reporter plasmids together with pRL-TK. Luciferase activity was measured with the Promega Dual-Luciferases Reporter Assay kit (Promega E1980) according to manufacturer's instructions. The relative firefly luciferase activity was normalized to renilla luciferase activity.

**Chromatin immunoprecipitation (ChIP)**. ChIP assay was performed using the Simple ChIP Enzymatic Chromatin IP Kit (Cell Signaling 9003) according to the manufacturer's protocol. The primers used to amplify the β-catenin target genes' promoter are shown in Supplementary Table 3.

**Quantitative real-time reverse-transcription PCR**. Total RNA was isolated using RNeasy reagents (Qiagen, Hilden, Germany) and then transcribed into complementary DNA using SuperScript III reagents (Invitrogen) with an oligo(dT)$_{20}$ primer. Quantitative real-time reverse-transcription PCR were performed using the SYBR green reagent on an 7900HT Real-Time PCR System (Applied Biosystems). The housekeeping gene, β-Actin, was used as a loading control. The sequences of primer sets are shown in Supplementary Table 4.

**The in vivo ubiquitylation assay**. H1299 or HEK293 cells were co-transfected with FBXW2, His-HA-Ub, and β-catenin, β-catenin-3A or β-catenin-3D, along with mock vector or FBXW2-ΔF controls. Cells were lysed in buffer A (6 M guanidine-HCl, 0.1 M Na2HPO4/NaH2PO4, and 10 mM imidazole, pH 8.0) and sonicated. The lysates were incubated with nickel-nitrilotriacetic acid (Ni-NTA) beads (QIAGEN) for 4 h at room temperature. The beads were washed twice with buffer A, twice with buffer A/TI (1 volume buffer A and 3 volumes buffer TI), and once with buffer TI (25 mM Tris-HCl and 20 mM imidazole, pH 6.8). The beads were boiled and the pull-down proteins were resolved by SDS-PAGE for immunoblotting using anti-β-catenin or anti-FLAG Ab.

**The in vitro ubiquitylation assay**. HA-FBXW2 or HA- FBXW2-ΔF was respectively purified from HEK293 cells transfected with either plasmid using HA beads (Sigma), and then eluted with $3 \times$ HA peptide (Sigma), serving as the source of E3. FLAG-tagged β-catenin, β-catenin-3A, or β-catenin-3D was respectively pulled down by FLAG beads (Sigma) from HEK293 cells transfected with according plasmid, serving as the substrate. The reaction was carried out at 37 °C for 1 h in 30 μl reaction buffer (40 mM Tris-HCl, pH 7.5, 2 mM ATP, 2 mM DTT, 5 mM MgCl$_2$) in the presence of E1, E2 and purified E3s. Poly-ubiquitinated β-catenin was resolved by SDS-PAGE, followed by IB with anti-FLAG Ab.

**Half-life analysis**. After gene manipulation, 20 μg/ml cycloheximide (CHX, Sigma) was added to the cell medium. At the indicated time points, cells were harvested, lysed and subjected to western blotting analysis. The densitometry quantification was performed using Image J processing software.

**Wound-healing assay**. The monolayer cells in 6-well plate were scraped in a straight line with a 10-μl pipette tip to produce a wound. Plate was then washed with PBS to remove detached cells and the cells were incubated in serum free medium. Photographs of the scratch were taken at 0 h, 24 h, or 48 h after wounding using Nikon inverted microscope. Gap width at 0 h was set to 1. Gap width analysis was performed with Image-Pro Plus 6.0 software. Multiple defined sites along the scratch were measured. Each scratch was given an average of all measurements. Data are shown as the average of three independent experiments.

**Migration and invasion assay**. For cell migration, $2–5\times10^4$ cells in 100 μl serum free medium were plated in an 8.0-mm, 24-well plate chamber insert (Corning Life Sciences, catalog no. 354578), with medium containing 10% fetal bovine serum (FBS) at the bottom of the insert. Cells were incubated for 12–24 h, and then fixed with 4% paraformaldehyde for 5 min. After washing with PBS for three times, cells were stained with 0.5% crystal violet blue for 5 min, and then washed with double-distilled H$_2$O. Cells on the upper surface of the insert were removed with a cotton swab. The positively stained cells were examined under the microscope. For the cell invasion assay, Matrigel-coated chambers (Corning Life Sciences, catalog no. 354483) were used instead of the chamber inserts used in migration assay.

**In vivo metastasis assay**. Five- to six-week-old BALB/c athymic nude mice (nu/nu, female) were purchased from Shanghai SLAC Laboratory Animal Center. Mice were fed with a regular diet (RD) and had free access to water and food. All mice procedures were approved by the Zhejiang University Committee on Use and Care of Animals. H1299 cells ($2\times10^6$) or H358 cells ($4\times10^6$) stably expressing indicated sh-RNA were injected via tail-vein. Five mice were used in each experimental group. After 6 or 8 weeks, mice were killed and all the lung tissues were collected, fixed and sectioned, followed by hematoxylin and eosin staining. Macroscopic metastases were quantified by counting lesions in all five lobes of the lung per mouse.

**Immunohistochemical staining and image analysis**. Human lung tumor tissue microarrays with lymph nodes metastasis data were purchased from Shanghai Outdo Biotech, China. For immune-histochemical staining, the sections were deparaffinized in xylene and rehydrated through graded ethanol. Antigen retrieval was performed for 20 min at 95 °C with 0.1% sodium citrate buffer (pH 6.0). Following quenching of endogenous peroxidase activity with 3% H$_2$O$_2$·dH$_2$O and blocking of non-specific binding with 1% bovine serum albumin buffer, sections were incubated overnight at 4 °C with an anti-β-catenin (Santa Cruz, SC-59737,1:400 dilution) or anti-FBXW2 (Proteintech, #11499-1-AP,1:100 dilution) antibody. Following several washes, the sections were treated with HRP conjugated secondary antibody for 30 min at room temperature, and stained with 0.05% 3, 3-diaminobenzidine tetrahydrochloride (DAB). Slides were photographed with Virtual slide microscope (Olympus VS120, Japan). The photographs were analyzed with the Image-Pro Plus 6.0 software (Media Cybernetics, Inc., Silver Spring, MD, USA).

Quantification of the immune-histochemical staining was conducted based on the ratio and intensity of the staining. Three random fields in each section were calculated. The measurement parameters collected by Image-Pro Plus 6.0 included density mean, area sum, and IOD. The optical density was counted and the area of

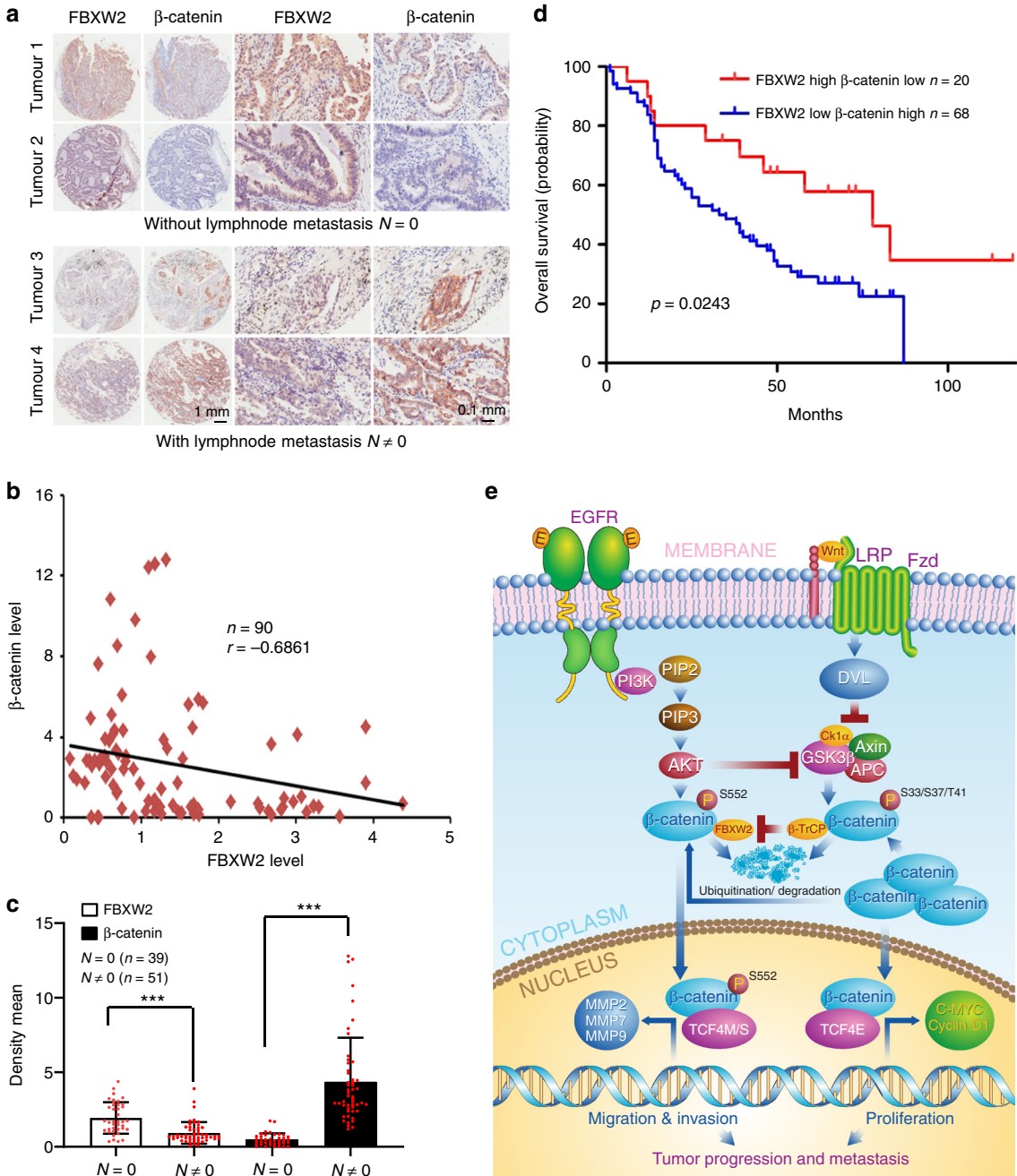

**Fig. 7** The levels of FBXW2 and β-catenin predict patient survival. **a**, **b** A general tendency of inversed protein levels between FBXW2 and β-catenin in lung tumors: Human lung cancer tissue microarrays containing 90 human lung cancer specimens were immune-stained with anti-FBXW2 and anti-β-catenin antibodies. Representative staining pictures of tumors are shown (**a**). Quantification of the staining intensity was performed by semi-quantitative scoring with all 90 tumors ($r = -0.6861$ $p < 0.001$, Pearson correlation coefficient) (**b**). **c** The expression levels of FBXW2 are reduced, while the expression levels of β-catenin are increased in the tumors with lymph-node metastasis: The staining intensity of FBXW2 and β-catenin in tumors with or without lymph-node metastasis ($N = 0$: without lymph-node metastasis, $N \neq 0$: with lymph-node metastasis) were statistically analyzed. Data are shown as mean ± s.d. ***$p < 0.001$ (Student's $t$-test). **d** The association of FBXW2 and β-catenin expression with overall survival in lung cancer patients. A total of 180 lung cancer tissue microarray samples were stained for FBXW2 and β-catenin levels. The Kaplan-Meier analysis was performed in 88 cancer samples showing an inverse correlation. The results showed that lower levels of FBXW2, coupled with higher levels of β-catenin, predict a worse overall patient survival ($p = 0.0243$, log-rank test). **e** Working model. EGF activation of EGFR leads to activation of AKT1, which inactivates GSK-3β to abrogate its binding with β-TrCP, but phosphorylates β-catenin at Ser552 to facilitate its binding with FBXW2 for subsequent ubiquitylation and degradation. In human cancer (e.g. lung cancer) with low level of FBXW2, phospho-β-catenin[ser552] is not degraded and translocated, along with non-degraded wild type β-catenin, to the nucleus, and forms a complex with TCF4M/S or TCF4E, leading to transactivation of the genes encoding MMPs or c-Myc/Cyclin D1 for enhanced migration/ invasion and proliferation, respectively (see text for more details)

interest was set through: hue, 0–30; intensity, 0–255, saturation, 0–255; next the image was converted to gray scale image, and the values were counted. The mean of density, equal to (IOD SUM)/area, represented the levels of specific protein per unit area.

**Statistical analysis**. The significance of the data between two experimental groups was determined by Student's $t$-test, and multiple group comparisons were analyzed by one-way ANOVA. All statistical analyses were two-sided, and different cut off values, $p < 0.05$ (*), $p < 0.01$ (**) and $p < 0.001$ (***), were considered significant.

**Reporting Summary**. Further information on experimental design is available in the Nature Research Reporting Summary linked to this article.

## Data availability
The authors declare that all the other data supporting the findings of this study are available within the paper and its supplementary information files, and from the corresponding author upon reasonable request. Gel source images for Figs. 1–4, 6 and Supplementary Figures 1-4, 7 are available in Supplementary Figure 9. A reporting summary for this Article is available as a Supplementary Information file.

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

## Acknowledgements

This work is supported in part by the National Key R&D Program of China (2016YFA0501800) (Y.S.), by the NCI grants CA156744 and CA171277 (Y.S.), and by the Chinese NSFC Grants 81572718 and 81630076 (Y.S.). We also thank Drs. Li Ma and Wenyi Wei for providing us various plasmid constructs.

## Author contributions

Conception and design: F.Y., X.X., and Y.S. Experiment execution: F.Y. Data acquisition: F.Y., J.X., H.L., and M.T. Data analysis and interpretation: F.Y. and Y.S. Manuscript writing: F.Y., X.X., and Y.S. Study supervision: Y.S.

## Additional information

**Competing interests:** The authors declare no competing interests.

