## [Peer Review File · Nature Communications]

Reviewers' comments:

Reviewer #1, Expertise: Ubiquitin (Remarks to the Author):

FBXW2 suppresses migration and invasion of lung cancer cells via promoting ubiquitylation and degradation of β -catenin

Fei Yang et al.

Yang, Sun, and colleagues provide several lines of evidence that FBXW2 regulates beta-catenin independent of canonical Wnt signaling, and that the CRL1-FBXW2 ligase complex plays a major role in RTK-AKT1-regulated beta-catenin stabilization. The role of AKT activity in potentiating beta-catenin signaling has been well-documented and the current manuscript reveals some potentially exciting findings that add to our understanding of the links between RTK/AKT and b-catenin signaling pathways. The current data are intriguing but preliminary and should be explored in more detail in order to strengthen the claims. The authors also need to more convincingly disentangle Wnt canonical vs. AKT-driven beta-catenin signaling in the investigated experimental systems.

Fig. 2: the authors should: 1) blot for b-TrCP and pS33/pS37-b-catenin with shFBXW2 and 2) knock down/express bTrCP and evaluate pS33/pS37- and pS552 b-catenin to evaluate potential cross-talk between the CRL1 ligase complexes and b-catenin phospho-substrates.

Lines 178-180, and other CHX time course experiments throughout (especially Fig. 4i): The effects of shFBXW2 or FBXW2 overexpression do not appear to be remarkable/significant at the indicated time points in Fig. 3a-c. In general, the authors should extend the treatment durations in order to evaluate whether longer treatments reveal more pronounced differences in b-cat degradation. If the experimental time frames are limited due to CHX toxicity, the authors should perform pulse-chase or similar experiments that are not complicated by the secondary effects of CHX toxicity.

Fig. 3c: FBXW2 overexpression markedly decreases total b-cat. Is the decrease specific to b-cat S552? Again the authors should evaluate bTrCP and S33/pS37-b-cat levels.

Lines 255-259: The authors should confirm that FBXW2 RNAi, FBXW2 ectopic expression, and GM6001 regulate MMP expression and that increased MMP expression regulates cell migration in the indicated models. The current experiments do not provide enough data to support the claims that FBXW2 expression regulates cell migration via b-cat-mediated MMP expression.

Some of the initial data (Fig 2d) support canonical TCF activity but this is less evident in other parts of the manuscript (Fig 6), where distinct beta-catenin activated forms have a different transcriptional output (MMPS vs Myc/proliferation). Is AKT activated beta-catenin still binding to TCF and relocalizing the complex to different chromatin-accessible regions? The authors should provide some additional mechanistic data to address these discrepancies.

It will also be important to evaluate the role of interfering with Wnt/beta-catenin signaling in these models. Porcupine inhibitors that prevent secretion of all Wnt ligands are readily accessible, also for in vivo experiments. The authors should perform such experiments to complement for example Fig 5c. In the current form, the rescue is performed by depleting the entire pool of beta-catenin. Notably, interfering with porcupine in murine NSCLC has been shown to alter tumor growth (Tammela et al, nature 2017).

The tail vein implantation model doesn't faithfully recapitulate tumor invasion and migration but rather initiation in the lung. This should be reflected more conservatively in the statements about invasion.

The Patient correlation is not very informative in the present form. What additional variables are

being included in that analysis? The authors should evaluate how Axin2 expression (or any other Wnt canonical target) tracks with b-catenin level and FBXW2 in the patient set.

Minor comments:

1. The term E3 ubiquitin ligase is redundant, the authors should use either the term E3 or ubiquitin ligase but not both. See <https://www.nobelprize.org/uploads/2018/06/ciechanover-lecture.pdf>.
2. Line 278: "These processes were..."

Reviewer #2, Expertise: Wnt (Remarks to the Author):

The manuscript by Yang and colleagues identifies beta-catenin as a novel substrate of the F-box E3 ubiquitin ligase FBXW2. In previous work, the Sun laboratory identified beta-catenin as a potential substrate of FBXW2 via affinity purification-mass spectrometry. In this study, the authors employ several biochemical methods, including immunoprecipitation, ubiquitylation assays as well as TOPflash reporter assays to demonstrate that FBXW2 targets beta-catenin for ubiquitylation and degradation.

Experiments indicate that FBXW2 forms a complex with beta-catenin which is enhanced upon beta-catenin phosphorylation at S552. FBXW2 binding to beta-catenin results in a reduction of protein levels (total and phosphorylated) and transcriptional activity of beta-catenin due to K48-linked polyubiquitylation. FBXW2-mediated beta-catenin degradation is EGF/AKT1 dependent but appears to be independent of the canonical Wnt/GSK3beta/beta-TRCP1 pathway.

The biological relevance of these novel findings was demonstrated by the observation that, in FBXW2-depleted cells, EGF-activated AKT1 promotes accumulation of pS552-modified beta-catenin, driving increased migration and invasive behavior of cells. The authors further show an inhibitory effect of FBXW2 on migration and invasion of lung cancer cells both in vitro and in vivo. Using a CHIP-based promoter-binding assay they were able to show that pS552 beta-catenin preferentially binds to the promoters of invasion/metastasis-promoting genes encoding the metalloproteases 2, 7 and 9. For clinical relevance, the authors found an inversed expression pattern between FBXW2 and beta-catenin in a tissue microarray from lung cancer patients, with a significant reduction in FBXW2 expression in tumors with lymph-node metastasis. In summary, a model is proposed in which the FBXW2-beta-catenin axis plays a critical role in the regulation of migration and invasion (metastatic behavior) of human lung cancer cells.

Overall, this is a well-focused and solid study that provides an interesting novel view on the role of beta-catenin in the regulation of cell migration and metastasis. I have a number of concerns that would need to be addressed to improve the manuscript.

Concerns:

1. Figure 2f: results would be strengthened if the authors add an immunoblot of endogenous beta-catenin. Furthermore, it is unclear what is meant with the *** significance line.
2. Figure 3a/b: FBXW2 knockdown extends the half-life of total beta-catenin as well as that of phosphorylated beta-catenin S552. Because of the stabilization of the total beta-catenin pool, stabilization of beta-catenin S552 could in fact be an indirect effect. The authors should demonstrate that S552A beta-catenin is more stable in EGF-AKT stimulated cells as compared to S552D beta-catenin. In addition, another phosphorylated form of beta-catenin should be checked (such as S675, PKA site) to validate these results.
3. Phosphorylation of beta-catenin by AKT seems to be the trigger for FBXW2-mediated ubiquitylation, however this pool of phosphorylated beta-catenin can also go into the nucleus to promote transcription of specific target genes. It is unclear how the authors envision this in a physiological situation. Does phosphorylated beta-catenin escape FBXW2 mediated ubiquitylation to travel to the nucleus? Or does this only happen in situations where FBXW2 is inhibited in some way or where AKT activity is high enough that it phosphorylates beta-catenin faster than FBXW2 can ubiquitylate it? Maybe the authors can speculate a bit more on this. Is FBXW2 specifically downregulated in migrating cells?

4. Figure 7d: The model should be explained in more detail. There are several points of convergence between the EGFR and Wnt/beta-catenin pathways which should be discussed in more depth and in light of the existing literature.
5. Figure 4h: what would happen when EGF treatment is combined with siGSK3beta or Wnt3a treatment with siAKT? Does EGF only stimulate phosphorylation of beta-catenin by AKT or does it also influence the canonical Wnt signaling? What is the effect of EGF treatment in a TOP-flash assay?
6. Line 213: AKT inhibition by MK2206 increased the level of β -catenin in figure 4c, but not in Figure 4b, can the authors explain this discrepancy?
7. Figure 7: Does the inverse expression pattern between FBXW2 and beta-catenin in lung cancer tissues impact patient survival? Kaplan Meier survival analysis would be informative.

Minor points:

1. Although the authors present some inverse correlations of beta-catenin and FBXW2 levels in lung cancer, the findings of their study would predict that FBXW2 (inactivating) mutations are likely to occur in cancer, is this the case?
2. Grammar in both the introduction and discussion could be improved. Particularly, lines 376-391.
3. Acronyms should be listed clearly e.g. SKP2 (line 24), AKT1 (line 89)
4. Line 79 In various cancers, including NSCLC.....please list references
5. Line 278 'was' should be 'were'

Reviewer #3, Expertise: lung cancer (Remarks to the Author):

This manuscript delineates a key role for the F-box E3 FBXW2 in the degradation of b-catenin in response to non-canonical b-catenin activators like EGFR signaling. They convincingly show that FBXW2 exhibits specific binding to b-cat under physiological conditions, dependent on S552 phosphorylation. They further show that b-cat protein levels, nuclear accumulation, and transcriptional activity are regulated by FBXW2 (increased by its knockdown and decreased by ectopic expression). These manipulations had the corresponding effects on b-cat protein half-life, dependent on S552 phosphorylation (the leveraging of both cKO and shRNA mediated elimination of FBXW2 is a strength). The critical role of FBXW2 in ubiquitin-mediated degradation of b-cat, via S552 phosphorylation and the K48 linkage of Ub, is further substantiated by in vitro and in vivo ubiquitination assays. The interaction between FBXW2 and b-cat is enhanced by EGFR signaling via AKT, which is required for S552 phosphorylation, FBXW2 interaction, and b-cat degradation. A real strength is that they show (comprehensively) how FBXW2 and b-TrCP1 dependent regulation of b-cat differ, with the former regulated by EGF/AKT and the latter by Wnt ligand/GSK3b. With minor concerns described below, results are clear and well controlled, and shown in multiple cell lines with diverse methods.

They go on to show that FBXW2 suppresses lung cancer migration and invasion, through degradation of b-cat. These in vitro studies are complemented with an in vivo lung metastasis mouse model (FBXW2 suppression increased metastases, which was reversed by b-cat knockdown). Interestingly, the phospho-mimetic form (552D) is most active at promoting migration/invasion and at increasing the expression of associated genes (MMPs), but the 33Y form (resistant to b-TrCP1 mediated degradation) is most active at promoting the expression of pro-proliferative genes (Myc and Cyclin D1). FBXW2 also preferentially regulates MMPs. This is really interesting, and adds new layers to the complexity of b-cat dependent regulation of different hallmarks of cancer. ChIP results substantiate these trends, in terms of how b-cat mutants and manipulating FBXW2 influences binding of b-cat to target promoters, although the ChIP results are not as clear cut as other assays (but the results are the results). It is also interesting that the

same site, 552, for which phosphorylation is important for b-cat dependent activation of MMP genes also mediates the degradation of b-cat via FBXW2 (presumably to down-modulate the response).

The relevance of their findings is further enhanced by the demonstration that FBXW2 and b-cat protein expression inversely correlate in human lung cancers, and lower FBXW2 or higher b-cat predicted increased lymph node metastases.

The differential regulation of invasive and proliferative features of b-cat by FBXW2 and b-TrCP1 are fascinating and important, particularly given the often-discussed (but to my knowledge not mechanistically substantiated) idea that migration and proliferation represent a mutual tradeoff. To migrate more entails proliferating less. So it would make sense that there would be a mechanism to toggle between these, and the authors have discovered such a mechanism involving a common transcription factor. Moreover, they show how downregulation of FBXW2 in cancers can boost the migratory impact of b-cat (perhaps without preventing its pro-proliferative roles).

The one area that I thought was left unexplained was how FBXW2 normally modulates EGF/EGFR dependent activation of b-cat and consequent migration and invasion, relative to proliferation. Most of the results were with EGFR wild-type lung cancers (like H1299). H358 was used for some studies, and it is EGFR-dependent. Performing additional manipulation of the FBXW2/b-cat axis in EGFR-dependent lung cancers (perhaps comparing those with activating EGFR mutations to those without) would put their results into the context of EGFR activation – how do these manipulations differentially impact these phenotypes in the cells with or without EGFR dependency? Still, while they write about the new mechanism as being one dependent on EGFR, it's not clear that it really is (i.e. it could be downstream of multiple drivers) – and EGFR-dependency is not really the point of their work. Given the depth that the authors went into uncovering this novel and important new mechanism of b-cat dependent transcriptional regulation of invasive properties, this and other important questions could be left for future studies. If so, the authors should modify how they present this new mechanism in terms of specificity for EGFR signaling (although specificity for AKT was nicely shown).

In all, this is a comprehensive study that reveals new roles for FBXW2 in regulating specific features of b-cat dependent transcription, leading to specific effects on migration/invasion/metastases which are distinct from the mechanism and impact of the previously described E3 ligase b-TrCP1. Only minor deficiencies were evident.

Issues:

- 1) For protein half-life determinations, the quantitations appear to be from single experiments with "n=1". The robustness of these determinations requires that repetitions be quantified and incorporated into half-life determinations (with statistics).
- 2) In Fig 6C, why does an MMP inhibitor decrease the expression of MMP mRNAs? There may be something that I don't know, but this wasn't clear to me.
- 3) The authors need to confirm that all error shown, and all statistics, were based on biological (not technical) replicates.

Minor:

- 1) Line 133: "constitutively active forms of β -catenin-S552D and β -catenin-3D"; I would refer to these as "phospho-mimetic", as these forms would actually be expected to be less "active" in terms of stability (albeit more active in terms of promoting transcription of some genes and migration/invasion, which we don't learn about till later in the manuscript). And I wouldn't refer to the S552A form as "inactive" for the same reason. True, that based on data presented later, the "active" designation makes sense, but I found it confusing when I read it early in the manuscript.

Thank you for your positive comments on our manuscript and your thoughtful critiques. Below, we have addressed point-by-point your constructive critiques, and modified the text accordingly.

Responses to Reviewer #1: (Expertise: Ubiquitin):

FBXW2 suppresses migration and invasion of lung cancer cells via promoting ubiquitylation and degradation of β -catenin

Fei Yang et al.

Yang, Sun, and colleagues provide several lines of evidence that FBXW2 regulates beta-catenin independent of canonical Wnt signaling, and that the CRL1-FBXW2 ligase complex plays a major role in RTK-AKT1-regulated beta-catenin stabilization. The role of AKT activity in potentiating beta-catenin signaling has been well-documented and the current manuscript reveals some potentially exciting findings that add to our understanding of the links between RTK/AKT and b-catenin signaling pathways. The current data are intriguing but preliminary and should be explored in more detail in order to strengthen the claims. The authors also need to more convincingly disentangle Wnt canonical vs. AKT-driven beta-catenin signaling in the investigated experimental systems.

Response: We thank the reviewer for his/her overall positive comments on our manuscript. We have addressed below all the critiques raised “to more convincingly disentangle Wnt canonical vs. AKT-driven beta-catenin signaling...”

Fig. 2: the authors should: 1) blot for b-TrCP and pS33/pS37-b-catenin with shFBXW2 and 2) knock down/express bTrCP and evaluate pS33/pS37- and pS552 b-catenin to evaluate potential cross-talk between the CRL1 ligase complexes and b-catenin phospho-substrates.

Response: Per reviewer's suggestions, we measured the protein levels of phosphorylated β -catenin-Ser33/37 and β -TRCP protein upon FBXW2 manipulations, and found that FBXW2 negatively regulates the levels of total β -catenin as well as phosphorylated β -catenin-Ser552, but had no effect on phosphorylated β -catenin-Ser33/37. On the other hand, β -TrCP1 manipulations negatively regulated protein levels of endogenous total β -catenin, phosphorylated β -catenin-Ser33/37 as well as FBXW2, but positively regulated phosphorylated β -catenin-Ser552. These results are consistent with our recent findings that β -TrCP1 negatively regulated FBXW2 by promoting its ubiquitylation and degradation (*Xu et al., 2017, Nat Commun 8:14002 | DOI: 10.1038/ncomms14002*). The newly generated data are now shown in Fig. 2a, b and Supplementary Fig. 2d, e.

Lines 178-180, and other CHX time course experiments throughout (especially Fig. 4i): The effects of shFBWX2 or FBXW2 overexpression do not appear to be remarkable/significant at the indicated time points in Fig. 3a-c. In general, the authors should extend the treatment durations in order to evaluate whether longer treatments reveal more pronounced differences in b-cat degradation. If the experimental time frames are limited due to CHX toxicity, the authors should perform pulse-chase or similar experiments that are not complicated by the secondary effects of CHX toxicity.

Response: We have performed suggested experiments by extending CHX treatment time up to 32 hrs. Indeed, more pronounced differences in both β -catenin and phosphorylated β -catenin-Ser⁵⁵² levels upon FBXW2 manipulation were observed. These newly generated data were now shown in Fig. 3a-c, 4i, and Supplementary Fig. 3e.

Fig. 3c: FBXW2 overexpression markedly decreases total b-cat. Is the decrease specific to b-cat S552? Again the authors should evaluate bTrCP and S33/pS37-b-cat levels.

Response: We performed this suggested experiment and found that FBXW2 ectopic expression significantly shortened the protein half-lives of both phosphor- β -cateninSer⁵⁵² and total β -catenin, but had no effect on those of β -TrCP1 and phosphor- β -cateninSer^{33/37} (Fig.3c).

Lines 255-259: The authors should confirm that FBXW2 RNAi, FBXW2 ectopic expression, and GM6001 regulate MMP expression and that increased MMP expression regulates cell migration in the indicated models. The current experiments do not provide enough data to support the claims that FBXW2 expression regulates cell migration via b-cat-mediated MMP expression.

Response: Per reviewer's suggestion, we have performed these experiments to determine the effects on migration/invasion upon MMPs knockdown alone or in combination with FBXW2 knockdown. The newly generated data (Supplementary Fig. 6a&b) showed that knockdown of MMP2 or MMP9 blocks cell migration/invasion, and significantly inhibits the migration/invasion induced by FBXW2 knockdown, indicating a causal role of MMPs, particularly MMP9, in siFBXW2-induced migration/invasion.

Some of the initial data (Fig 2d) support canonical TCF activity but this is less evident in other parts of the manuscript (Fig 6), where distinct beta-catenin activated forms have a different transcriptional output (MMPS vs Myc/proliferation). Is AKT activated beta-catenin still binding

to TCF and relocalizing the complex to different chromatin-accessible regions? The authors should provide some additional mechanistic data to address these discrepancies.

Response: We appreciated this excellent comment and have performed suggested experiment by transfecting four FLAG-tagged plasmids expressing 1) β -catenin-WT, 2) β -catenin-S33Y (resistant to β -TrCP mediated degradation), 3) β -catenin-S552A (resistant to FBXW2-mediated degradation), and 4) β -catenin-S552D (AKT-activated form), along with 5) the vector control, followed by FLAG-IP and TCF4-IB. As shown in newly generated data (Supplementary Fig. 7c), β -catenin-S33Y and β -catenin-S552D differentially bind to different isoforms of TCF4 (TCF4E and TCF4M/S, respectively). It has been reported previously that different TCF-4 isoforms generated by alternative splicing events activated different downstream target genes in hepatocellular carcinoma (HCC) and renal cell carcinoma (RCC) (*Andreas et al., 2010, Nucleic acids research, Vol. 38, No. 6 | DOI: 10.1093/nar/gkp1197, Yoshito et al., 2013, Liver Int: 33: 1100–1112 | DOI:10.1111/liv.12188, Hiroaki et al., 2003, Clinical Cancer Research, Vol. 9: 2121-2132*). This preferred binding with different TCF4 isoforms by β -catenin-S33Y and β -catenin-S552D would likely facilitate differential activation of c-Myc/cyclinD vs. MMPs. We have now discussed this possibility in the Discussion section (page 19).

It will also be important to evaluate the role of interfering with Wnt/beta-catenin signaling in these models. Porcupine inhibitors that prevent secretion of all Wnt ligands are readily accessible, also for in vivo experiments. The authors should perform such experiments to complement for example Fig 5c. In the current form, the rescue is performed by depleting the entire pool of beta-catenin. Notably, interfering with porcupine in murine NSCLC has been shown to alter tumor growth (Tammela et al, nature 2017).

Response: This is an excellent suggestion and thank you. We have now performed this suggested experiment using cell culture setting. The results showed that porcupine inhibitor LGK941 (at 1 and 5 μ M), which sufficiently blocks the secretion of all Wnt ligands (Tammela et al, nature 2017) only partially and moderately blocks (in a dose dependent manner) cell migration and invasion stimulated by FBXW2 knockdown (Supplementary Fig. 6c), suggesting that unlike the Wnt- β -TrCP1- β -catenin axis, which mainly controls the proliferation, the AKT1-FBXW2- β -catenin axis plays the major role in cell migration and invasion.

With regards to the use of this compound (in replacement of sh- β -catenin) for the *in vivo* tail-vein injection experiment, we feel that the results will not be informative and conclusive. This reviewer correctly pointed out that “*The tail vein implantation model doesn’t faithfully recapitulate tumor invasion and migration but rather initiation in the lung*”. Given the Wnt signal inhibitor would certainly inhibit growth of tumor cells after initiation in the lung, we feel that this *in vivo* experiment will not be able to distinguish the role played by the Wnt- β TrCP- β -catenin axis (for proliferation) from that by the AKT-FBXW2- β -catenin-S552 (for migration and invasion). We sincerely hope that the reviewer will agree with us.

The tail vein implantation model doesn’t faithfully recapitulate tumor invasion and migration but rather initiation in the lung. This should be reflected more conservatively in the statements about invasion.

Response: Agreed. We have now changed the statement to a more conservative one to reflect this concern.

The Patient correlation is not very informative in the present form. What additional variables are being included in that analysis? The authors should evaluate how Axin2 expression (or any other Wnt canonical target) tracks with b-catenin level and FBXW2 in the patient set.

Response: The goal of this experiment is to find correlation between β -catenin and FBXW2. We agree that it will be nice to include a correlation study between Axin2 (as Wnt canonical target) and FBXW2, but it is out of the focus of this study, thus not absolutely necessary. Furthermore, we purchased human lung cancer tissue microarray long time ago and are NOT able to get the consecutive sections from the same microarray for this Axin2 staining. So we decided not perform this “nice-to-have” type of experiment. Instead we preformed the Kaplan Meier survival analysis, as suggested by reviewer 2, to determine whether inversed expression pattern between FBXW2 and β -catenin impacts patient survival with inclusion of additional tumor tissue microarray samples, and found that their inverse correlation indeed significantly impacts the patient survival (Fig. 7d).

Minor comments:

1. *The term E3 ubiquitin ligase is redundant, the authors should use either the term E3 or ubiquitin ligase but not both. See <https://www.nobelprize.org/uploads/2018/06/ciechanover-lecture.pdf>.*

Response: Thanks. It has been corrected.

2. *Line 278: “These processes were...”*

Response: It has been corrected. Thanks.

Responses to Reviewer #2, Expertise: Wnt:

The manuscript by Yang and colleagues identifies beta-catenin as a novel substrate of the F-box E3 ubiquitin ligase FBXW2. In previous work, the Sun laboratory identified beta-catenin as a potential substrate of FBXW2 via affinity purification-mass spectrometry. In this study, the authors employ several biochemical methods, including immunoprecipitation, ubiquitylation assays as well as TOPflash reporter assays to demonstrate that FBXW2 targets beta-catenin for ubiquitylation and degradation.

Experiments indicate that FBXW2 forms a complex with beta-catenin which is enhanced upon beta-catenin phosphorylation at S552. FBXW2 binding to beta-catenin results in a reduction of protein levels (total and phosphorylated) and transcriptional activity of beta-catenin due to K48-linked polyubiquitylation. FBXW2-mediated beta-catenin degradation is EGF/AKT1 dependent but appears to be independent of the canonical Wnt/GSK3beta/beta-TRCP1 pathway.

The biological relevance of these novel findings was demonstrated by the observation that, in FBXW2-depleted cells, EGF-activated AKT1 promotes accumulation of pS552-modified beta-

catenin, driving increased migration and invasive behavior of cells. The authors further show an inhibitory effect of FBXW2 on migration and invasion of lung cancer cells both in vitro and in vivo. Using a CHIP-based promoter-binding assay they were able to show that pS552 beta-catenin preferentially binds to the promoters of invasion/metastasis-promoting genes encoding the metalloproteases 2, 7 and 9. For clinical relevance, the authors found an inversed expression pattern between FBXW2 and beta-catenin in a tissue microarray from lung cancer patients, with a significant reduction in FBXW2 expression in tumors with lymph-node metastasis. In summary, a model is proposed in which the FBXW2-beta-catenin axis plays a critical role in the regulation of migration and invasion (metastatic behavior) of human lung cancer cells.

Overall, this is a well-focused and solid study that provides an interesting novel view on the role of beta-catenin in the regulation of cell migration and metastasis. I have a number of concerns that would need to be addressed to improve the manuscript.

We thank the reviewer for his/her overall positive comments on our manuscript. We have addressed below all the critiques this reviewer raised.

Concerns:

*1. Figure 2f: results would be strengthened if the authors add an immunoblot of endogenous beta-catenin. Furthermore, it is unclear what is meant with the *** significance line.*

Response: Per reviewer's suggestion, we have added the blot detecting endogenous β -catenin protein, and found that it is reduced by HA-FBXW2, not by HA-FBXW2- Δ F (Fig. 2f). The *** represents $p < 0.001$ (One-way ANOVA).

2. Figure 3a/b: FBXW2 knockdown extends the half-life of total beta-catenin as well as that of phosphorylated beta-catenin S552. Because of the stabilization of the total beta-catenin pool, stabilization of beta-catenin S552 could in fact be an indirect effect. The authors should demonstrate that S552A beta-catenin is more stable in EGF-AKT stimulated cells as compared to S552D beta-catenin. In addition, another phosphorylated form of beta-catenin should be checked (such as S675, PKA site) to validate these results.

Response: Per reviewer's suggestion, we measured the protein half-life of ectopically expressed β -catenin-WT, β -catenin-S33Y, β -catenin-S552D, β -catenin-S552A in EGF-AKT stimulated cells and found that β -catenin-S552A has the longest protein half-life (Supplementary Fig. 3e). We focused on phosphorylation of beta-Catenin on S552 and S33, not on S675, because the former two sites mediate beta-Catenin ubiquitylation and degradation by FBXW2 and beta-TrCP, respectively.

3. Phosphorylation of beta-catenin by AKT seems to be the trigger for FBXW2-mediated ubiquitylation, however this pool of phosphorylated beta-catenin can also go into the nucleus to promote transcription of specific target genes. It is unclear how the authors envision this in a physiological situation. Does phosphorylated beta-catenin escape FBXW2 mediated ubiquitination to travel to the nucleus? Or does this only happen in situations where FBXW2 is inhibited in some way or where AKT activity is high enough that it phosphorylates beta-catenin faster than FBXW2 can ubiquitylate it? Maybe the authors can speculate a bit more on this. Is FBXW2 specifically downregulated in migrating cells?

Response: The reviewer raised an excellent point! Indeed, it is very likely that under the situation when AKT1 is activated (e.g. by EGF), while FBXW2 is downregulated (often seen in cancer tissues), cancer cells would obtain an increased capacity of migration and invasion. We have added this point in our Discussion section (page 20).

With regards to the question “*Is FBXW2 specifically downregulated in migrating cells?*”, We feel that this is a correlation type of study and may not be very informative, given that many genetic alterations occur in different cancer cell lines. Nevertheless, we have convincingly showed in our paired siFBXW2 vs. siCtrl experiments in multiple cancer lines that FBXW2 knockdown indeed promotes migration.

4. *Figure 7d: The model should be explained in more detail. There are several points of convergence between the EGFR and Wnt/beta-catenin pathways which should be discussed in more depth and in light of the existing literature.*

Response: As kindly suggested by the reviewer, we have now provided more detailed description of the model in figure legends of Fig. 7e.

5. *Figure 4h: what would happen when EGF treatment is combined with siGSK3beta or Wnt3a treatment with siAKT? Does EGF only stimulate phosphorylation of beta-catenin by AKT or does it also influence the canonical Wnt signaling? What is the effect of EGF treatment in a TOP-flash assay?*

Response: We performed the IP assay and found the β -catenin-FBXW2 interaction can be enhanced by EGF treatment regardless of GSK3 β conditions. On the other hand, β -catenin- β -TrCP1 interaction was inhibited by Wnt3a treatment regardless of AKT1 conditions (Fig. 4h). As shown in Fig. 4k, EGF treatment activated the transcriptional activity of β -catenin which can be inhibited by FBXW2, whereas wnt3a enhanced transcriptional activity cannot be inhibited by FBXW2.

6. *Line 213: AKT inhibition by MK2206 increased the level of β -catenin in figure 4c, but not in Figure 4b, can the authors explain this discrepancy?*

Response: Difference is due to the inclusion or exclusion of MG132. As kindly suggested by the reviewer, we have now marked it in the figure by including one line for MG132 (Fig. 4b).

7. *Figure 7: Does the inverse expression pattern between FBXW2 and beta-catenin in lung cancer tissues impact patient survival? Kaplan Meier survival analysis would be informative.*

Response: We thank the reviewer for this excellent suggestion. We have previously shown that FBXW2 expression is down regulated in lung cancer tissues and its downregulation correlated with poor survival of lung cancer patients (n=102) (Xu et al, 2017, Nat Commun 8:14002 | DOI: 10.1038/ncomms14002). Consistently, in the revised manuscript, we performed the Kaplan Meier survival analysis with inclusion of additional lung cancer tissue microarray samples. Among 88 cancer samples with inverse correlation between FBXW2 and β -catenin, we found that high FBXW2/low β -catenin (n=20) predicts a better patient survival, while low

FBXW2/high β -catenin (n=68) was associated with poor survival of lung cancer patients (Fig. 7d). Thus, inverse expression pattern between FBXW2 and beta-catenin in lung cancer tissues indeed impacts the patient survival.

Minor points:

1. *Although the authors present some inverse correlations of beta-catenin and FBXW2 levels in lung cancer, the findings of their study would predict that FBXW2 (inactivating) mutations are likely to occur in cancer, is this the case?*

Response: Yes, indeed, we have recently shown FBXW2 is mutated in various human cancers. We characterized one loss-of-function mutant (S84C), and one gain-of-function mutant (E269K) (Xu et al., 2017, Nat Commun 8:14002 | DOI: 10.1038/ncomms14002).

2. *Grammar in both the introduction and discussion could be improved. Particularly, lines 376-391.*

Response: The grammar has been improved in these sections, including indicated paragraph.

3. *Acronyms should be listed clearly e.g. SKP2 (line 24), AKT1 (line 89)*

Response: We have added these.

4. *Line 79 In various cancers, including NSCLC.....please list references*

Response: The references have been added.

5. *Line 278 'was' should be 'were'*

Response: It has been corrected. Thanks.

Reviewer #3, Expertise: lung cancer:

This manuscript delineates a key role for the F-box E3 FBXW2 in the degradation of b-catenin in response to non-canonical b-catenin activators like EGFR signaling. They convincingly show that FBXW2 exhibits specific binding to b-cat under physiological conditions, dependent on S552 phosphorylation. They further show that b-cat protein levels, nuclear accumulation, and transcriptional activity are regulated by FBXW2 (increased by its knockdown and decreased by ectopic expression). These manipulations had the corresponding effects on b-cat protein half-life, dependent on S552 phosphorylation (the leveraging of both cKO and shRNA mediated elimination of FBXW2 is a strength). The critical role of FBXW2 in ubiquitin-mediated degradation of b-cat, via S552 phosphorylation and the K48 linkage of Ub, is further substantiated by in vitro and in vivo ubiquitinylation assays. The interaction between FBXW2 and b-cat is enhanced by EGFR signaling via AKT, which is required for S552 phosphorylation, FBXW2 interaction, and b-cat degradation. A real strength is that they show (comprehensively) how FBXW2 and b-TrCP1 dependent regulation of b-cat differ, with the

former regulated by EGF/AKT and the latter by Wnt ligand/GSK3b. With minor concerns described below, results are clear and well controlled, and shown in multiple cell lines with diverse methods.

They go on to show that FBXW2 suppresses lung cancer migration and invasion, through degradation of b-cat. These in vitro studies are complemented with an in vivo lung metastasis mouse model (FBXW2 suppression increased metastases, which was reversed by b-cat knockdown). Interestingly, the phospo-mimetic form (552D) is most active at promoting migration/invasion and at increasing the expression of associated genes (MMPs), but the 33Y form (resistant to b-TrCP1 mediated degradation) is most active at promoting the expression of pro-proliferative genes (Myc and Cyclin D1). FBXW2 also preferentially regulates MMPs. This is really interesting, and adds new layers to the complexity of b-cat dependent regulation of different hallmarks of cancer. ChIP results substantiate these trends, in terms of how b-cat mutants and manipulating FBXW2 influences binding of b-cat to target promoters, although the ChIP results are not as clear cut as other assays (but the results are the results). It is also interesting that the same site, 552, for which phosphorylation is important for b-cat dependent activation of MMP genes also mediates the degradation of b-cat via FBXW2 (presumably to down-modulate the response).

The relevance of their findings is further enhanced by the demonstration that FBXW2 and b-cat protein expression inversely correlate in human lung cancers, and lower FBXW2 or higher b-cat predicted increased lymph node metastases.

The differential regulation of invasive and proliferative features of b-cat by FBXW2 and b-TrCP1 are fascinating and important, particularly given the often-discussed (but to my knowledge not mechanistically substantiated) idea that migration and proliferation represent a mutual tradeoff. To migrate more entails proliferating less. So it would make sense that there would be a mechanism to toggle between these, and the authors have discovered such a mechanism involving a common transcription factor. Moreover, they show how downregulation of FBXW2 in cancers can boost the migratory impact of b-cat (perhaps without preventing its pro-proliferative roles).

Response: We thank the reviewer for his/her excellent summary of our study, and very positive comments on our manuscript.

The one area that I thought was left unexplained was how FBXW2 normally modulates EGF/EGFR dependent activation of b-cat and consequent migration and invasion, relative to proliferation. Most of the results were with EGFR wild-type lung cancers (like H1299). H358 was used for some studies, and it is EGFR-dependent. Performing additional manipulation of the FBXW2/b-cat axis in EGFR-dependent lung cancers (perhaps comparing those with activating EGFR mutations to those without) would put their results into the context of EGFR activation – how do these manipulations differentially impact these phenotypes in the cells with or without EGFR dependency? Still, while they write about the new mechanism as being one dependent on EGFR, it's not clear that it really is (i.e. it could be downstream of multiple drivers) – and EGFR-dependency is not really the point of their work. Given the depth that the authors went into uncovering this novel and important new mechanism of b-cat dependent transcriptional regulation of invasive properties, this and other important questions could be left for future studies. If so, the authors should modify how they present this new

mechanism in terms of specificity for EGFR signaling (although specificity for AKT was nicely shown).

Response: We value reviewer's comment and agreed with the reviewer that "...EGFR-dependency is not really the point of" our work, but it is an interesting point for future investigation. Nevertheless, we indeed showed that the EGF-AKT signal controls beta-catenin degradation by FBXW2. We have now modified language with more focus on AKT, rather than upstream EGFR throughout the manuscript.

In all, this is a comprehensive study that reveals new roles for FBXW2 in regulating specific features of b-cat dependent transcription, leading to specific effects on migration/invasion/metastases which are distinct from the mechanism and impact of the previously described E3 ligase b-TrCP1. Only minor deficiencies were evident.

Response: Again, thank you for your appreciation of our work. We have addressed below all the critiques raised.

Issues:

1) For protein halflife determinations, the quantitations appear to be from single experiments with "n=1". The robustness of these determinations requires that repetitions be quantified and incorporated into halflife determinations (with statistics).

Response: As kindly suggested by the reviewer, we repeated these experiments and performed Statistical analysis shown in Fig. 3a-c, 4d, 4i.

2) In Fig 6C, why does an MMP inhibitor decrease the expression of MMP mRNAs? There may be something that I don't know, but this wasn't clear to me.

Response: It is reported that GM6001 is a broad spectrum inhibitor of MMPs. It binds to the critical active-site zinc atom of this class of proteinases to inhibit their activities. Interestingly, it also inhibits MMPs' transcription in some tumors (*Li et al., 2017, Oncotarget, Vol. 8, No. 37: 60789-60808/ DOI: 10.18632/oncotarget.18487, Chen et al., 2015, Molcular Cancer 14:83/ DOI: 10.1186/s12943-015-0348-7*).

3) The authors need to confirm that all error shown, and all statistics, were based on biological (not technical) replicates.

Response: Yes, we stated "shown is mean \pm SEM from three independent experiments" and the methods of statistical analysis in the figure legends.

Minor:

1) Line 133: "constitutively active forms of β -catenin-S552D and β -catenin-3D"; I would refer to these as "phospho-mimetic", as these forms would actually be expected to be less "active" in terms of stability (albeit more active in terms of promoting transcription of some genes and migration/invasion, which we don't learn about till later in the manuscript). And I wouldn't refer to the S552A form as "inactive" for the same reason. True, that based on data presented later,

the “active” designation makes sense, but I found it confusing when I read it early in the manuscript.

Response: Per reviewer’s suggestion, we have now changed these descriptions to avoid the confusion.

We feel that our manuscript has been significantly strengthened after these modifications, and thank you again for your thoughtful critiques.

Sincerely yours,

Yi Sun,
Professor of Radiation Oncology

REVIEWERS' COMMENTS:

Reviewer #1 (Remarks to the Author):

Most comments have been adequately addressed. However there remain two outstanding issues that need to be corrected in order to recommend publication.

Point #1

Original reviewer comment: It will also be important to evaluate the role of interfering with Wnt/beta-catenin signaling in these models. Porcupine inhibitors that prevent secretion of all Wnt ligands are readily accessible, also for in vivo experiments. The authors should perform such experiments to

complement for example Fig 5c. In the current form, the rescue is performed by depleting the entire pool of beta-catenin. Notably, interfering with porcupine in murine NSCLC has been shown to alter tumor growth (Tammela et al, nature 2017).

Author response: This is an excellent suggestion and thank you. We have now performed this suggested experiment using cell culture setting. The results showed that porcupine inhibitor LGK941 (at 1 and 5 μ M), which sufficiently blocks the secretion of all Wnt ligands (Tammela et al, nature 2017) only partially and moderately blocks (in a dose dependent manner) cell migration and invasion stimulated by FBXW2 knockdown (Supplementary Fig. 6c), suggesting that unlike the Wnt-b-TrCP1-b-catenin axis, which mainly controls the proliferation, the AKT1-FBXW2-b-catenin axis plays the major role in cell migration and invasion.

With regards to the use of this compound (in replacement of sh-b-catinin) for the in vivo tail-vein injection experiment, we feel that the results will not be informative and conclusive. This reviewer correctly pointed out that "The tail vein implantation model doesn't faithfully recapitulate tumor invasion and migration but rather initiation in the lung". Given the Wnt signal inhibitor would certainly inhibit growth of tumor cells after initiation in the lung, we feel that this in vivo experiment will not be able to distinguish the role played by the Wnt-bTrCP-b-catinin axis (for proliferation) from that by the AKT-FBXW2-b-catinin-S552 (for migration and invasion). We sincerely hope that the reviewer will agree with us.

Reviewer response: On the contrary, these data would be very informative. If no rescue of LGK is observed in vivo, then the current observation made in this manuscript will be strengthened. If rescue is indeed observed, it would caution the authors to be more conservative with their conclusions.

Point #2

Original reviewer question: The tail vein implantation model doesn't faithfully recapitulate tumor invasion and migration but rather initiation in the lung. This should be reflected more conservatively in the statements about invasion.

Author response: Agreed. We have now changed the statement to a more conservative one to reflect this concern.

Reviewer response: Please clarify where the statement has been changed. It should be acknowledged as a limitation of the study. In the present form, there is no data on actual spontaneous metastasis.

Reviewer #2 (Remarks to the Author):

The authors have addressed all my points appropriately - either discussing them in the text or adding new experimental data.

Reviewer #3 (Remarks to the Author):

The authors have done a thorough job of addressing reviewer concerns, and the manuscript is now even stronger. The greater attention to demonstrating reproducibility is appreciated, as is the greater insight into the underlying mechanism provided by the new experiments. The only question that I have concerns their interpretation of the results in Supplemental Figure 6c with LGK 974 – the data shows a pretty clear rescue of the increased migration mediated by FbxW2 knockdown using LGK, but they interpret this rescue as “modest”. I think that they should reconsider their interpretation. Perhaps the canonical b-TrCP1-dependent pathway for stabilization of b-catenin is still important, which shouldn’t detract from the FbxW2 regulated pathway that is more specific for invasion/metastasis.

There were a couple of very minor items that could be quickly corrected (all by editing):

1. In multiple places, they spell “phospho-” as “phosphor-”. This is the fault of Word auto-correction.
2. Line 440: “VRK2-mediated phosphorylation” should be corrected.

Signed - James DeGregori

Responses to REVIEWERS' COMMENTS:

Reviewer #1 (Remarks to the Author):

Most comments have been adequately addressed. However there remain two outstanding issues that need to be corrected in order to recommend publication.

Point #1

Original reviewer comment: It will also be important to evaluate the role of interfering with Wnt/beta-catenin signaling in these models. Porcupine inhibitors that prevent secretion of all Wnt ligands are readily accessible, also for in vivo experiments. The authors should perform such experiments to complement for example Fig 5c. In the current form, the rescue is performed by depleting the entire pool of beta-catenin. Notably, interfering with porcupine in murine NSCLC has been shown to alter tumor growth (Tammela et al, nature 2017).

Author response: This is an excellent suggestion and thank you. We have now performed this suggested experiment using cell culture setting. The results showed that porcupine inhibitor LGK941 (at 1 and 5 μ M), which sufficiently blocks the secretion of all Wnt ligands (Tammela et al, nature 2017) only partially and moderately blocks (in a dose dependent manner) cell migration and invasion stimulated by FBXW2 knockdown (Supplementary Fig. 6c), suggesting that unlike the Wnt-b-TrCP1-b-catinin axis, which mainly controls the proliferation, the AKT1-FBXW2-b-catinin axis plays the major role in cell migration and invasion.

With regards to the use of this compound (in replacement of sh-b-catinin) for the in vivo tail-vein injection experiment, we feel that the results will not be informative and conclusive. This reviewer correctly pointed out that “The tail vein implantation model doesn’t faithfully recapitulate tumor invasion and migration but rather initiation in the lung”. Given the Wnt signal inhibitor would certainly inhibit growth of tumor cells after initiation in the lung, we feel that this in vivo experiment will not be able to distinguish the role played by the Wnt-bTrCP-b-catinin axis (for proliferation) from that by the AKT-FBXW2-b-catinin-S552 (for migration and invasion). We sincerely hope that the reviewer will agree with us.

Reviewer response: On the contrary, these data would be very informative. If no rescue of LGK is observed in vivo, then the current observation made in this manuscript will be strengthened. If rescue is indeed observed, it would caution the authors to be more conservative with their conclusions.

We have discussed this issue with editors before the submission of our revision. Editors have now agreed that we do not need to perform this *in vivo* experiment. Thank you!

Point #2

Original reviewer question: The tail vein implantation model doesn’t faithfully recapitulate tumor invasion and migration but rather initiation in the lung. This should be reflected more

conservatively in the statements about invasion.

Author response: Agreed. We have now changed the statement to a more conservative one to reflect this concern.

Reviewer response: Please clarify where the statement has been changed. It should be acknowledged as a limitation of the study. In the present form, there is no data on actual spontaneous metastasis.

We agreed with the reviewer, and have now stated in the Results section that “by targeting β -catenin, FBXW2 acts as an inhibitory protein against migration, invasion and metastasis of lung cancer cells in both *in vitro* cell culture and *in vivo* mouse models, although the tail-vein injection of tumor cells is not a spontaneous metastasis model.” (lines 290-291).

Reviewer #2 (Remarks to the Author):

The authors have addressed all my points appropriately - either discussing them in the text or adding new experimental data.

This reviewer has no further issue. Thank you.

Reviewer #3 (Remarks to the Author):

The authors have done a thorough job of addressing reviewer concerns, and the manuscript is now even stronger. The greater attention to demonstrating reproducibility is appreciated, as is the greater insight into the underlying mechanism provided by the new experiments. The only question that I have concerns their interpretation of the results in Supplemental Figure 6c with LGK 974 – the data shows a pretty clear rescue of the increased migration mediated by FbxW2 knockdown using LGK, but they interpret this rescue as “modest”. I think that they should reconsider their interpretation. Perhaps the canonical b-TrCP1-dependent pathway for stabilization of b-catenin is still important, which shouldn't detract from the FbxW2 regulated pathway that is more specific for invasion/metastasis.

Thank you. Now we have provided more detailed description of Figure S6C, and as follows:
“The results showed that LGK941 at 1 μ M had very minimal, if any, effect, while at 5 μ M, it showed approximately 50% of blockage cell migration and invasion stimulated by FBXW2 knockdown (Supplementary Fig. 6c), suggesting that the Wnt- β -TrCP1- β -catenin axis, while mainly controls proliferation, also contributes to migration and invasion, whereas the AKT1-FBXW2- β -catenin axis is more specific for migration and invasion.” (lines 276-281)

There were a couple of very minor items that could be quickly corrected (all by editing):

- 1. In multiple places, they spell “phospho-“ as “phosphor-”. This is the fault of Word auto-correction.*
- 2. Line 440: “VRK2-mediated phosphorylation” should be corrected.*

Thank you. We have now made all suggested corrections.